# *FoxG1* regulates the formation of cortical GABAergic circuit during an early postnatal critical period resulting in autism spectrum disorder-like phenotypes

Goichi Miyoshi [1✉], Yoshifumi Ueta[1], Akiyo Natsubori [2], Kou Hiraga [1], Hironobu Osaki [1], Yuki Yagasaki[1], Yusuke Kishi [3], Yuchio Yanagawa[4], Gord Fishell[5,6,7], Robert P. Machold[5] & Mariko Miyata [1]

Abnormalities in GABAergic inhibitory circuits have been implicated in the aetiology of autism spectrum disorder (ASD). ASD is caused by genetic and environmental factors. Several genes have been associated with syndromic forms of ASD, including *FOXG1*. However, when and how dysregulation of *FOXG1* can result in defects in inhibitory circuit development and ASD-like social impairments is unclear. Here, we show that increased or decreased *FoxG1* expression in both excitatory and inhibitory neurons results in ASD-related circuit and social behavior deficits in our mouse models. We observe that the second postnatal week is the critical period when regulation of *FoxG1* expression is required to prevent subsequent ASD-like social impairments. Transplantation of GABAergic precursor cells prior to this critical period and reduction in GABAergic tone via *Gad2* mutation ameliorates and exacerbates circuit functionality and social behavioral defects, respectively. Our results provide mechanistic insight into the developmental timing of inhibitory circuit formation underlying ASD-like phenotypes in mouse models.

[1] Department of Neurophysiology, Tokyo Women's Medical University, Shinjuku, Tokyo, Japan. [2] Sleep Disorders Project, Tokyo Metropolitan Institute of Medical Science, Setagaya, Tokyo, Japan. [3] Graduate School of Pharmaceutical Sciences, University of Tokyo, Tokyo, Japan. [4] Department of Genetic and Behavioral Neuroscience, Gunma University Graduate School of Medicine, Maebashi, Japan. [5] NYU Neuroscience Institute, Smilow Research Center, New York University School of Medicine, New York, NY, USA. [6] Department of Neurobiology, Blavatnik Institute, Harvard Medical School, Boston, MA, USA. [7] Stanley Center at the Broad Institute, Cambridge, MA, USA. ✉email: Goichi.Miyoshi@gmail.com

ASDs are characterized by social communication deficits and restricted/repetitive behavioral patterns (DSM-5), with a prevalence around 1–2% of all children (USA: CDC estimate). While the vast majority of ASDs are thought to emerge through a combination of genetic and environmental risk factors, a percentage (~5%) of ASD cases are syndromic, in that they are caused by highly penetrant mutations in a single gene, such as *MECP2* (Rett or duplication), *FMR1* (Fragile X) or *SHANK3* (Phelan-McDermid)[1]. Despite their relative rarity, these syndromic forms of ASD have provided key insights into the broader etiological basis of ASD. Of particular interest, many syndromic ASD models show an increase in the cortical synaptic excitation-inhibition (E/I) ratio, a phenomenon that has been hypothesized to underlie the cognitive and behavioral symptoms of ASD[2,3].

Inhibition mediated by cortical GABAergic interneurons is known to regulate animal social behaviors[4] and abnormal GABAergic signaling has been implicated in both schizophrenia and ASD[2,3,5,6]. Loss of inhibitory neurons has been observed in postmortem ASD patient studies[7] and consistent with this, children with ASD are more likely to have epilepsy. In ASD modeling studies, conditional mutations of syndromic ASD genes in GABAergic populations often reproduce the ASD phenotypes found in the respective null animals[8,9]. Developmentally, GABAergic systems are known to regulate circuit plasticity and are capable of accelerating or delaying the onset of the visual critical period[10]. Furthermore, transplantation of GABAergic neuronal precursors has been shown to induce plasticity in the mature visual cortex[11]. However, how GABAergic circuit dysfunction during development contributes to adult ASD phenotypes is still not well understood. Indeed, the elevated cortical excitatory-inhibitory ratio observed in syndromic ASD models has been recently reported not to reflect network hyperexcitability per se, but rather is a homeostatic compensation of synaptic drive in the adult circuits[12]. Thus, investigating how perturbations in inhibitory circuit development contribute to the etiology of ASD should help innovate therapeutic approaches.

Recently, *FOXG1*, a critical transcription factor for forebrain development[13–21] has been linked to ASD and other neurodevelopmental disorders such as schizophrenia[22]. Studies in patient-derived iPS cells have suggested that *FOXG1* dysregulation broadly contributes to the etiology of idiopathic ASD[23]. Furthermore, perturbation of *FOXG1* has been implicated in a new type of syndromic ASD (FOXG1 syndrome), with gene duplications (gain-of-function)[24,25] or loss-of-function mutations (heterozygous haploinsufficiency)[26–29] observed in affected patients[30–32]. Moreover, recent advances in sequencing techniques have revealed *FOXG1* point-mutations in idiopathic ASD children (FOXG1 Research Foundation), suggesting that there are more undiscovered FOXG1 syndrome patients world-wide. However, both when during development and in which brain circuits *FOXG1* dysregulation contributes to ASD etiology have not been elucidated.

Here we demonstrate that neuronal *FoxG1* dysregulation precipitates ASD-related social impairments in our mouse models when *FoxG1* expression changes are paired in both excitatory and inhibitory populations. We find that the second postnatal week is a critical period for developing ASD phenotypes upon *FoxG1* dysregulation. Indeed, at the end of this critical period, the cortical excitatory-inhibitory ratio is already imbalanced in all three variants of *FoxG1* Low and High ASD models. We show that the GABAergic system plays pivotal roles during the critical period by demonstrating that weakening or supplementing cortical GABAergic tone exacerbates or ameliorates the social behavioral impairments of the *FoxG1* ASD model, respectively. We thus identify both the critical time-window and inhibitory circuit mechanisms that lead to ASD-related social behavioral

impairments upon *FoxG1* dysregulation. Our results support the promise of therapeutic approaches to ameliorate ASD phenotypes through the early intervention of cortical GABAergic systems.

## Results

### A rodent FOXG1 syndrome model exhibiting ASD-like social behavioral impairments.
In order to address the developmental circuit mechanisms underlying the social behavioral alterations found in ASD patients, we first examined if ASD FOXG1 syndrome could be recapitulated in rodents to model the disease phenotypes. We utilized the three-chamber assay to test for ASD-related unconditioned social behaviors[4,8,33–37] in *FoxG1* Heterozygous adult animals (6 weeks old)[13]. In this behavioral assay, animals are first allowed to freely explore the three chambers connected by corridors (Fig. 1a, Habituation). Next, an age and gender-matched stranger wild type mouse is introduced in a cage located in one chamber, allowing the test animal to make a choice between exploring a chamber with a novel mouse versus an empty cage (Fig. 1a, Sociability). In the third session, preference of the test animal to explore the chamber with an already familiar mouse versus the chamber with a newly introduced stranger mouse is compared (Fig. 1a, Social Novelty). As expected, wild type animals spent a significantly increased amount of time in the social side of the chambers in both sociability (Fig. 1b, open orange bar) and social novelty assays (filled orange bar). However, *FoxG1* Heterozygous animals showed no obvious preference for the social side of the chamber (Fig. 1b, left versus right chamber) and instead spent a significantly increased amount of time in the center chamber (Fig. 1b, compare filled purple bars of Nov), suggesting decreased sociability, consistent with ASD-like social behavioral deficits. Indeed, the sociability score was found to be decreased in the *FoxG1* Heterozygous animals (Fig. 1c, scored based on the orange bar graphs in 1b).

We carried out a battery of additional behavioral analyses of this *FoxG1* Heterozygous mouse model to test whether there were other behavioral abnormalities beyond the impairments in contextual fear memory reported previously[38]. Briefly, while general motor functions of the Heterozygous animals were comparable to the controls (Fig. 1d), Heterozygous animals showed decreased anxiety (Fig. 1e) and impairments in working memory (Fig. 1f).

### Neither the loss of GABAergic cells nor microcephaly per se causes ASD social phenotypes in *FoxG1* haploinsufficient animals.
Haploinsufficient FOXG1 syndrome ASD patients exhibit microcephaly[27,28], and likewise, reduction in the size of the cerebral hemispheres has been reported in at least three different *FoxG1* Heterozygous mouse models[38–40] and was observed here (Fig. 2a, b). In addition, consistent with the cases reported in ASD postmortem studies[7], we further found a reduction in GABAergic neuron numbers within the medial prefrontal (prelimbic and infralimbic) and somatosensory barrel cortices in these animals (Fig. 2d–g, k). We hypothesized that the microcephaly phenotype is due to the role of *FoxG1* in progenitor proliferation within the developing forebrain[18,19,41,42]. We thus preserved *FoxG1* expression in progenitors (Fig. 2c) by utilizing *Cre* drivers for postmitotic glutamatergic (*Nex-Cre*)[43] and GABAergic (*Dlx-Cre*)[44] neurons of the forebrain and used them in conjunction with one copy of a conditional loss-of-function *FoxG1* allele[20] (Fig. 2l). Indeed, in this postmitotic neuronal Heterozygous model, both cortical thickness as well as GABAergic cell density were rescued to levels comparable to control animals in both medial prefrontal and somatosensory barrel cortices (Fig. 2h–k). Importantly, this neuronal postmitotic *FoxG1* haploinsufficiency

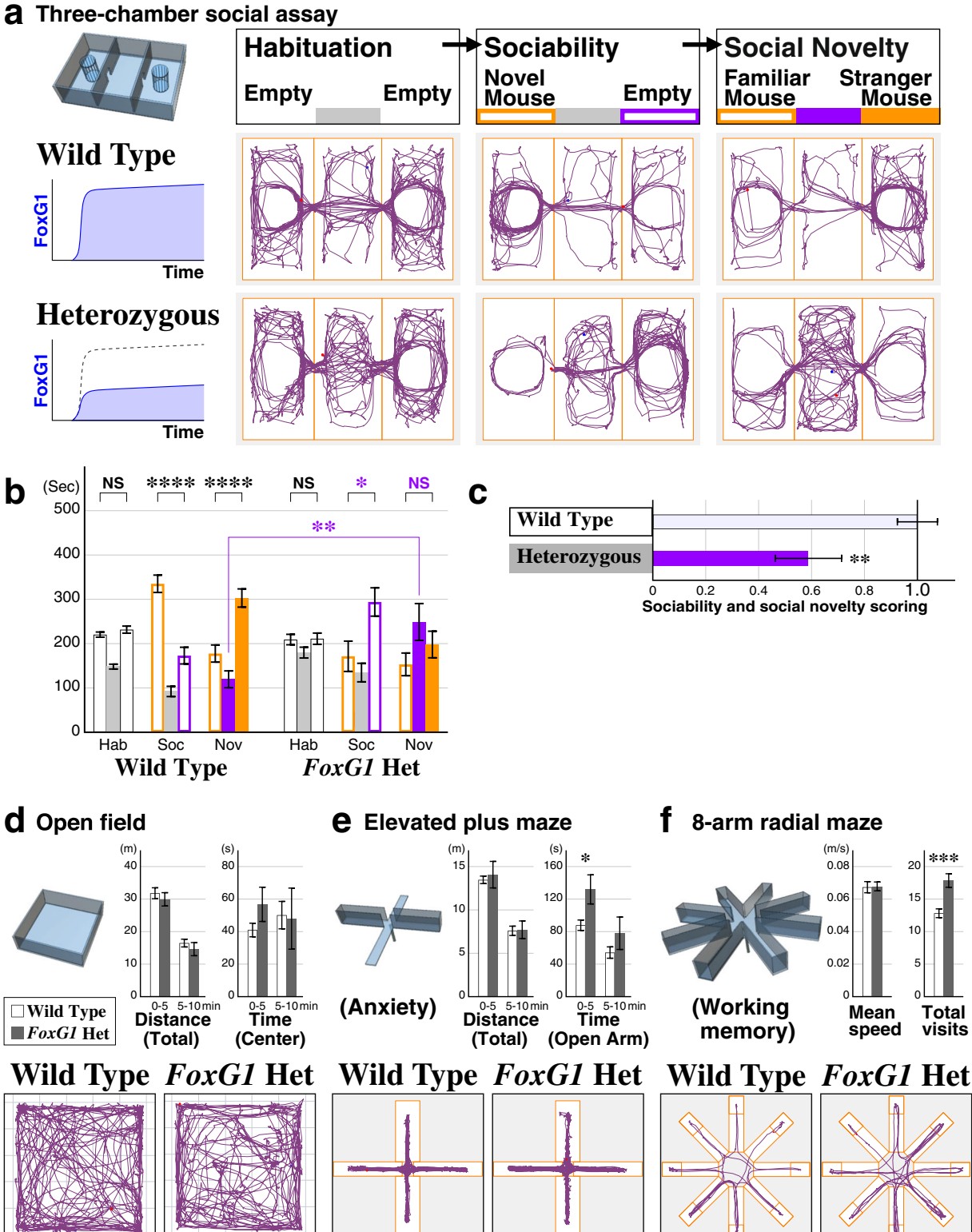

## Haploinsufficiency model

**a** Three-chamber social assay

model also spares glial *FoxG1* expression[45–47] unlike the Heterozygous model (Fig. 2c).

We first asked if neuronal *FoxG1* haploinsufficiency is sufficient to reproduce the ASD-phenotypes observed in *FoxG1* Heterozygous animals (Fig. 2m, Glu+GABA Low, traces in Supplementary Fig. 1). Indeed, *FoxG1* haploinsufficiency in both pyramidal and GABAergic neurons led to social behavioral

impairments in a manner comparable to the Heterozygous model (Fig. 1b). This suggests that neither microcephaly nor loss of GABAergic cells causes social behavioral impairments per se. We next asked if ASD-related social behavioral deficits originate from excitatory or inhibitory circuit mechanisms upon removal of one copy of *FoxG1*. Reduction of *FoxG1* levels in postmitotic pyramidal cells alone did not affect sociability (Fig. 2m, Glu

**Fig. 1 The FOXG1 syndrome Heterozygous model mouse displays ASD-like social behavioral impairments. a–c** Representative traces of animal location of wild type and *FoxG1* Heterozygous (*LacZ* knock-in) littermates are shown for each 10 min of the Habituation, Sociability, and Social Novelty sessions (**a**). Social behavior of the animals was analyzed by comparing the time spent in each chamber of the three-chamber assay (**b**). Social behavior scores of the animals are calculated based on the time spent at the social side (orange bars in **b**) of the chambers (**c**, $p = 0.00753$**). Wild type animals preferred to spend time in the social side of the chambers (**b**, orange bar graphs) while Heterozygous animals did not exhibit a social preference (**b**, purple * and NS for left vs. right chamber bar graphs). WT: $n = 26$, $p = 0.261$(Hab), $p = 2.55 \times 10^{-7}$****(Soc), $p = 4.90 \times 10^{-5}$**** (Nov), Het: $n = 26$, $p = 0.911$(Hab), $p = 0.0117$*(Soc), $p = 0.262$(Nov). In addition, Heterozygous animals preferred the middle chamber (filled purple bar graph, $p = 0.00759$**) in the third social novelty session and avoided the two lateral chambers harboring other mice. **d–f** Other behavioral paradigms tested: Heterozygous animals showed normal locomotion (**d**, $n = 14$ each), mildly decreased anxiety (**e**, $n = 14$ each, $p = 0.0334$*) and reduced working memory (**f**, $n = 28$ each, $p = 0.000180$***). Data are mean ± SEM, $p$ values are from two-tailed $t$-test.

Low). GABA Low littermate animals unexpectedly exhibited an increased preference for social novelty, with animals spending significantly more time in chambers containing mice in comparison to controls (Fig. 2m, GABA Low, orange asterisks). Thus, our results indicate that in *FOXG1* haploinsufficiency ASD patients, reduced sociability is due to *FOXG1* hypofunction in both excitatory and inhibitory postmitotic neurons.

**ASD phenotypes in haploinsufficient and duplication FOXG1 syndrome models arise from paired *FoxG1* changes in both excitatory and inhibitory neurons.** Increased *FOXG1* gene copy number can also result in ASD symptoms[24,25]; thus, we sought to examine how *FoxG1* gain-of-function could affect social behavior. Considering that *FoxG1* gain-of-function can disrupt the migration of pyramidal neurons[20], we chose to augment *FoxG1* levels during the early postnatal period after most neuronal migration is complete (~1st postnatal week; P7)[48]. We again utilized *Cre* drivers specific for excitatory and inhibitory lineages in the forebrain, here combined with a conditional transactivator line (*R26-stop-tTA*) and a transactivator-dependent *FoxG1* expression allele (*TRE-FoxG1*), and *FoxG1* gain-of-function was initiated only after P7 (Fig. 3a). Using the three-chamber assay, we found that animals exhibited social behavioral deficits only when *FoxG1* is augmented in both Glu+GABA lineages simultaneously (Fig. 3b, traces in Supplementary Fig. 2). This outcome in the sociability scores is similar to that observed upon reduction of *FoxG1* expression (Fig. 3c), albeit with a milder phenotype in social novelty sessions compared to the Heterozygous (Fig. 1b) or neuronal Glu+GABA Low models (Fig. 2m). In sum, we conclude that either up- or downregulation of *FoxG1* expression levels impairs social behavior, but only when the perturbation occurs in both excitatory and inhibitory populations (Fig. 3c, purple bars).

Upon establishing three ASD FOXG1 syndrome models of Heterozygous, Low (neuronal Glu + GABA Low) and High (1 week Glu + GABA High), in addition to the social behavioral evaluations, we cross-correlated other behavioral features and cortical EEG patterns. In contrast to the *FoxG1* Heterozygous animals (Fig. 1f), no working memory deficits were observed in *FoxG1* Low and High models (Fig. 3d, see also Supplementary Fig. 3). *FoxG1* Heterozygous animals are often aggressive, and attack siblings or partners during mating, but these behaviors were also not evident in the Low and High models. In addition, no clear anxiety phenotype was observed in the *FoxG1* Low and High models in the open field and elevated plus maze assays (Fig. 3e, f, see also Supplementary Figs. 1 and 2). These results indicate that working memory impairment, aggression or anxiety does not account for the social behavior abnormalities observed in our *FoxG1* Low and High perturbation models.

We next tested if the gamma-band frequency of EEG is attenuated in a manner similar to ASD patients[49,50] or in rodent models[4,51,52]. We performed EEG recordings during the awake free-moving state with the electrodes aligned nearby the medial

prefrontal cortex (Fig. 3g to j, Supplementary Fig. 4). In the control 8 weeks old wild type mice, the power spectrum peak in the high-frequency range was found nearby 58 Hz (Fig. 3k, orange trace). However, this peak was reduced as well as shifted to a lower frequency range in the Heterozygous model (Fig. 3k, black trace), with peak amplitude reduced in the Low model (Fig. 3k, gray trace), or the peak shifted to lower frequency in the High model (Fig. 3k, blue trace), resulting in overall reduction in high-gamma frequency power (55–80 Hz) in these socially impaired *FoxG1* models (Fig. 3l). While epilepsies have been observed in some human FOXG1 syndrome patients[24,26–28,30,31], we did not detect any obvious epileptic behavior in any of our *FoxG1* perturbation models.

In order to verify the expected alterations in FoxG1 expression in our socially impaired *FoxG1* mutant models, we measured the FoxG1 protein levels within the medial prefrontal cortex at postnatal 2 and 6 weeks by western blotting (Fig. 3m–o). By removing one copy of *FoxG1*, FoxG1 protein levels were reduced to approximately half of that in the control animals in both Heterozygous and Low models (Fig. 3o). This result indicates that a compensatory upregulation of *FoxG1* mRNA transcription likely does not occur in haploinsufficient *FOXG1* patients either. Already at postnatal 2 weeks, *FoxG1* High manipulation at 1 week was sufficient to increase FoxG1 protein levels to that observed in 6-week old animals (Fig. 3o). We further validated our gain-of-function method by confirming that the increase of FoxG1 is suppressed by continuous feeding of doxycycline (Fig. 3o, +Dox). We conclude that our genetic strategies enabled precise manipulations in FoxG1 protein levels in our model animals.

**An early juvenile ASD critical period identified through temporal *FoxG1* dysregulation and imbalanced excitatory-inhibitory circuits.** While excess copy number of *FOXG1* has been identified in syndromic forms of ASD, the ectopic upregulation of *FOXG1* expression has been suggested to be a key contributor to disease etiology in idiopathic ASD patients[23]. We thus took advantage of our doxycycline-repressible gain-of-function system (Fig. 3a) and carried out *FoxG1* High (Glu + GABA) manipulations during selective postnatal time-windows (Fig. 4a). When *FoxG1* augmentation is delayed for 2 weeks (Fig. 4a, 3w High), instead of 1 week (Figs. 4a and 3b), this no longer resulted in social behavioral impairments (Fig. 4a), demonstrating that *FoxG1* augmentation per se does not affect animal social behavior. Thus, we performed *FoxG1* augmentation exclusively between 1 and 3 weeks by replacing the doxycycline diet with a regular one only during this period and found that these animals (1–3w High) showed alterations in sociability to a similar extent as 1 week High animals (Fig. 4a, scores compared in Fig. 4b). We further carried out *FoxG1* augmentation between 1 and 2 weeks or 2–3 weeks (Fig. 4a, 1–2w and 2–3w High) and discovered that between 1 and 2 weeks is the critical time-window for *FoxG1* augmentation (Fig. 4b). Western blotting analysis revealed that the changes in FoxG1 protein levels occur within

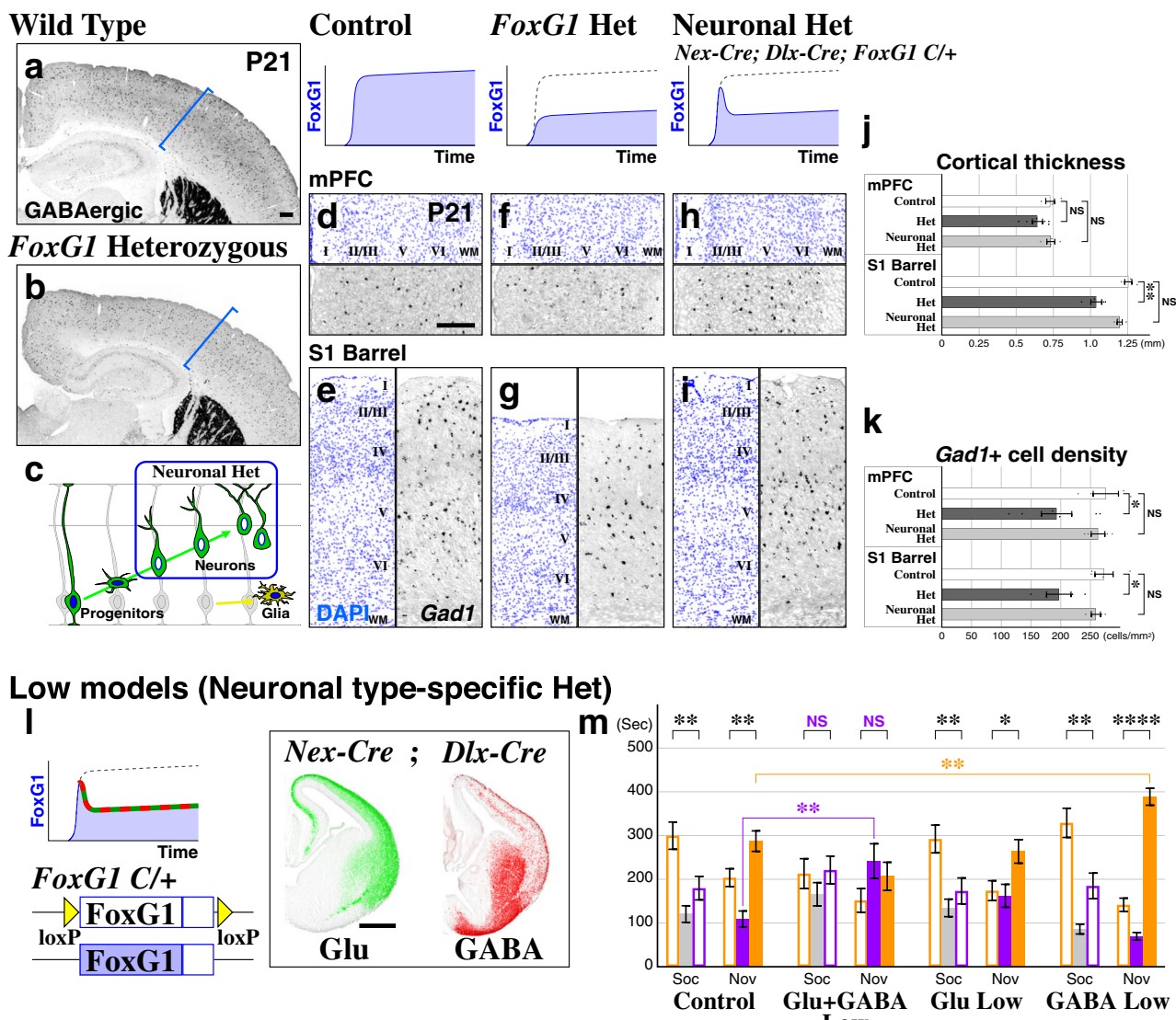

**Fig. 2 ASD FOXG1 syndrome social phenotypes are due to neuronal *FoxG1* haploinsufficiency, but not microcephaly or loss of GABAergic cells. a**, **b** Consistent with the microcephaly phenotypes found in human *FOXG1* haploinsufficiency patients, Heterozygous animals (**b**) exhibit a smaller brain size and reduced GABAergic cell numbers (labeled by *Dlx5a-Cre*; *Ai9*) compared to wild type (**a**). Blue brackets indicate the thickness of the barrel cortex in wild type animals. **c** In the neuronal *FoxG1* Heterozygous model, *FoxG1* haploinsufficiency only takes place in postmitotic neurons, thus sparing the progenitors and glia of the neocortex. **d**–**i** In situ hybridization with a *Gad1* antisense probe (black) labels GABAergic cell populations, with DAPI stain (blue) to visualize the laminar structure of the cortex in Control (**d**, **e**), *FoxG1* Heterozygous (**f**, **g**) and the postmitotic neuronal Heterozygous model (**h**, **i**: *Nex-Cre*; *Dlx-Cre*; *FoxG1-C/+*, scheme in **l**). In the Heterozygous model, the cortical thickness of the S1 Barrel (**g**) but not the mPFC (**f**) is reduced compared to the Control (**d**, **e**). Removal of one copy of *FoxG1* selectively in postmitotic neuronal populations rescues the Heterozygous microcephaly phenotype (**h**–**j**). While GABAergic cell density is reduced in both the mPFC and S1 Barrel cortex of the Heterozygous model (**f**, **g**) compared to the Control (**d**, **e**), this phenotype was completely rescued in Neuronal Het animals (**h**, **i**, **k**). **j**, **k** The thickness of cortical layers 1–6 (**j**) as well as *Gad1*-positive cell density (**k**) were analyzed. Control: $n = 4$, *FoxG1* Het: $n = 6$(mPFC) or 4(S1BF), Neuronal Het: $n = 4$, Control vs. Het: $p = 0.0945$ (**j**: mPFC), $= 0.00398$**(**j**: S1), $= 0.0377$* (**k**: mPFC), $= 0.0276$*(**k**: S1), Control vs. Neuronal Het: $p = 0.909$, $= 0.101$, $= 0.613$, $= 0.463$ (same order with Het). **l**, **m** Genetic strategy (**l**) and results from three-chamber social interaction experiments of *FoxG1* Low model (**m**). For the sake of clarity, in the bar graphs (**m**), the first 10 min of habituation data is not included (see the full data sets in Supplementary Fig. 1). **m** Neuronal Het animals without any microcephaly or GABAergic cell loss phenotypes showed ASD-like social behavioral impairments similar to the Heterozygous model (Fig. 1a–c). This was not the case in either Glu Low or GABA Low models and unexpectedly, GABA Low models showed a strong preference for the social side of the chamber (orange asterisks) during the social novelty session. Data are mean ± SEM, $p$ values are from two-tailed $t$-test.

2 days upon addition or removal of the doxycycline-containing diet (Supplementary Fig. 5d).

Upon identifying a critical period for *FoxG1* dysregulation, we hypothesized that there would be a shared circuit deficit found across all three *FoxG1* ASD models particularly during this time-window. To this end, we evaluated excitatory-inhibitory (E/I) balance by performing slice electrophysiological recordings of mEPSC and mIPSC events in layer 2/3 pyramidal neurons of the medial prefrontal cortex (prelimbic and infralimbic, Fig. 4c, right) in our three ASD FOXG1 syndrome models of Heterozygous, Low (neuronal Glu + GABA Low) and High (1 week Glu + GABA High). Upon examining 2-week old juvenile animals, we found that the E/I ratio within the medial prefrontal cortex was altered in all of these ASD models (Fig. 4d, e). However, to our

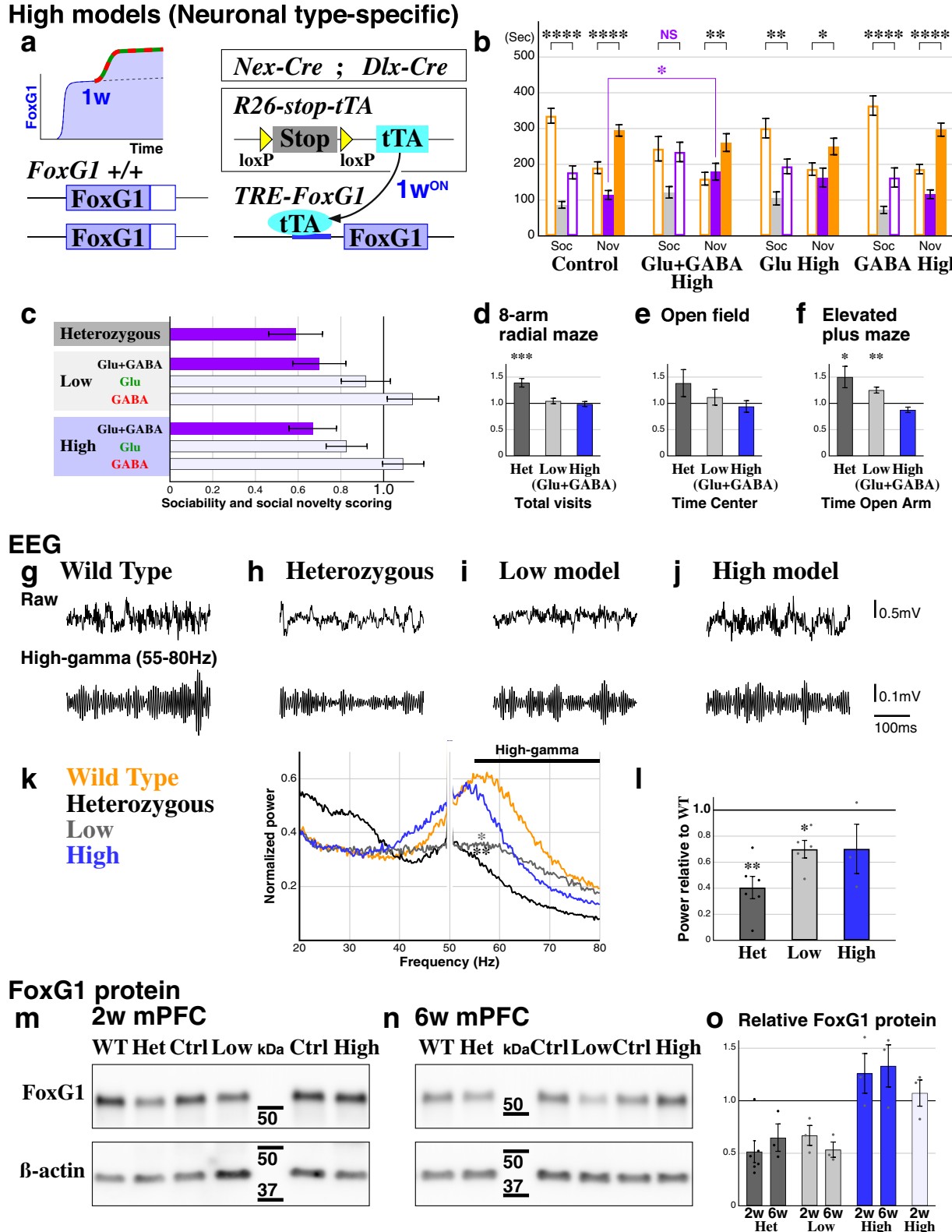

surprise, and distinct from other rodent ASD models[12], we found no obvious change in the E/I balance within the adult medial prefrontal cortex in our *FoxG1* ASD models (Fig. 4f–h). In fact, not just the E/I ratio but the values for the mEPSC and mIPSC were comparable to that of controls (Fig. 4g, h). The exception was a slight decrease in the E/I frequency ratio in the Heterozygous medial prefrontal cortex due to a reduced mEPSC frequency (Fig. 4g), although this was not the case in the barrel field (Supplementary Fig. 6). These data suggest that, in our *FoxG1* perturbation ASD models, although the transiently

**Fig. 3 ASD social phenotypes arise from a balanced decrease or increase of *FoxG1* in excitatory and inhibitory populations. a–c** A genetic strategy for the ligand-inducible *FoxG1* augmentation High model (**a**) and results from three-chamber social interaction experiments (**b**, **c** and see the full data sets in Supplementary Fig. 2). Only the Glu+GABA High model exhibited a clear social behavioral alteration, with milder phenotypes observed during the social novelty session (**b**). A summary diagram of the social behaviors of *FoxG1* Het (Fig. 1b), Low (Fig. 2m) and High (**b**) models (**c**). In order to cross-correlate the animal models for two different social assays, time spent in the novel side of the chamber during the sociability session (orange bar) was multiplied to the time spent outside the center chamber during the social novelty session (orange bars). The models with ASD-like social behavioral alterations are colored in purple. Data are normalized to the values of the controls within each group. **d–f** Other behavioral paradigms tested on the Low (Glu + GABA) and High (Glu + GABA) models. Unlike the Heterozygous animals (Fig. 1f), both Low ($n = 68$, control $n = 62$) and High ($n = 34$, control $n = 48$) models show no clear deficits in working memory (**d**). Both Low ($n = 34$, control $n = 31$) and High ($n = 17$, control $n = 24$) models show normal locomotion (**e**), and Low model displayed mildly decreased anxiety (**f**, $p = 0.00181**$), in a manner similar to the Heterozygous animals (Fig. 1e). Please see Supplementary Figs. 1–3 for the complete data analysis. **g–j** EEG recording was carried out in awake freely behaving adult (7–8 weeks) wild type ($n = 4$) and littermate *FoxG1* Heterozygous ($n = 6$) animals and was compared to Low ($n = 5$) and High models ($n = 3$). Representative examples of raw (top) and filtered high-gamma frequency traces (bottom). **k**, **l** Normalized power spectrum in medium to high-frequency ranges are shown (**k**). Note that power line noise at 50 Hz is masked. The peak at high-frequency range in the wild type (54.5–58.5 Hz) was significantly reduced in the Heterozygous and Low models ($p = 0.00568**$ and $= 0.0143*$). This peak was shifted toward low-frequency range in the High model (**k**). EEG high-gamma frequency power relative to the wild type animals is shown as bar graphs for the *FoxG1* Heterozygous ($p = 0.00254**$), Low ($p = 0.0466*$) and High models (**l**). See Supplementary Fig. 4 for the theta and low-gamma frequency analysis. **m–o** FoxG1 protein levels in the mPFC was analyzed in the *FoxG1* Heterozygous, Low and High models and compared to the control animals by using western blotting (**m**, **n**, see the full data sets in Supplementary Fig. 5). FoxG1 levels are reduced to approximately half in the Heterozygous and Low, and increased 1.3 times in the High model at postnatal 2 and 6 weeks (**o**). We confirmed that continuous doxycycline (Dox) feeding effectively suppresses FoxG1 augmentation. $n = 3$ each, except for 2w WT and Het $n = 6$. Data are mean ± SEM, $p$ values are from two-tailed $t$-test.

imbalanced E/I ratio within the juvenile medial prefrontal cortex normalizes by adult ages, this results in permanent circuit changes as indicated by altered gamma EEG power (Fig. 3l) and impaired social behavior (Fig. 3c).

**Decreased GABAergic activity exacerbates ASD social phenotypes in the FOXG1 syndrome model.** In the *FoxG1* Heterozygous model, the E/I ratio in the cortical circuit is transiently increased immediately after the critical period (Fig. 4d). We thus hypothesized that this model precipitates ASD phenotypes through a decrease in inhibitory GABAergic tone. In order to test this, we first addressed the roles of GABAergic transmission for the development of ASD phenotypes by analyzing *Gad2* mutant animals[53,54] in the three-chamber assay (Fig. 5a). In contrast to its isoform Gad1, Gad2 predominantly resides at presynaptic terminals to synthesize GABA locally. Thus, removal of the *Gad2* gene effectively reduces overall GABAergic synaptic tone and its loss has been shown to modify the visual critical period[10,55]. In the three-chamber sociability assay, while we found no obvious changes in the *Gad2* heterozygotes (Fig. 5b), *Gad2* nulls did not show a preference for the social side of the chamber (Fig. 5b, NS purple). During the social novelty session, although *Gad2* null animals still preferred the chamber with a novel animal, they spent more time in the center chamber (Fig. 5b, purple asterisks) and thus their overall sociability score was reduced (Fig. 5d). These data suggest that the reduction of GABAergic tone in *Gad2* nulls leads animals to develop ASD-like social behavioral impairments. Interestingly, the social behavior score of the *Gad2* nulls was decreased to a level comparable to the *FoxG1* Heterozygous model (Fig. 5d purple bars).

We next analyzed the consequence of reducing GABAergic transmission in the *FoxG1* Heterozygous background by cross-comparing the *Gad2* littermate animals (Fig. 5c, d). In the *FoxG1* Heterozygous background, both *Gad2* heterozygous and null animals showed no preference to the novel animal (Fig. 5c) and preferred the center chamber in the social novelty session (Fig. 5c), suggesting that they exhibit ASD-like social impairments. Moreover, both *Gad2* heterozygous and null mutations further reduced the sociability score of the *FoxG1* Heterozygotes (Fig. 5d dark bars). We further found that reduction of GABAergic tone does not affect the working memory of the

animals (Fig. 5e) in contrast to the sociability (Fig. 5d). These data suggest that reducing GABAergic tone exacerbates the social behavioral alterations found in the ASD *FoxG1* haploinsufficiency mouse model. This further supports the idea that the *FoxG1* Heterozygous model develops ASD phenotypes through alterations in GABAergic systems during the early juvenile critical period.

**Augmentation of GABAergic systems during the juvenile critical period re-balances the cortical E/I ratio and ameliorates the social impairments of the *FoxG1* ASD model.** In the FOXG1 syndrome haploinsufficiency model, cortical inhibition is reduced during the critical period (Fig. 4d, e), and further attenuation of GABAergic tone exacerbated the social impairments of this model (Fig. 5). We thus hypothesized that a proper E/I ratio during the early juvenile critical period is important for the development of ASD phenotypes, and that augmentation of GABAergic tone during this period might re-adjust the E/I imbalance and improve the behavioral outcomes of *FoxG1* Heterozygotes. In order to test this, we focused on the mPFC, a brain region implicated in social behavior[4], and we took advantage of the GABAergic cell precursor transplantation technique. This method has been shown in several contexts to therapeutically modify recipient mature neural circuits by transplanted donor cells forming new inhibitory synapses and creating circuit plasticity[11,56–58]. We dissociated EGFP-labeled GABAergic cell precursors (*Dlx-Cre*; *RCE:loxP*[59,60]) from the E13.5 medial ganglionic eminences (MGE) that are wild type for *FoxG1*, bilaterally transplanted these cells into the postnatal 1 week medial prefrontal cortices, and analyzed E/I ratios at 2 weeks of age (Fig. 6a, scheme). When the miniature frequencies of mPFC layer 2/3 cells were analyzed one week after the transplantation (Fig. 6b), we found that the increased E/I ratio in the Heterozygous animals was rescued to wild type levels upon receiving GABAergic neuron precursor transplantation (Fig. 6c, NS in orange). Consistent with this, GABAergic precursor transplantation did not attenuate the peak amplitude E/I ratio of the Heterozygous models (Fig. 6d), suggesting that GABAergic cell precursor transplantation at one week normalized the excitatory-inhibitory local circuit balance in the Heterozygous animal model to levels comparable to wild type. We further confirmed that the

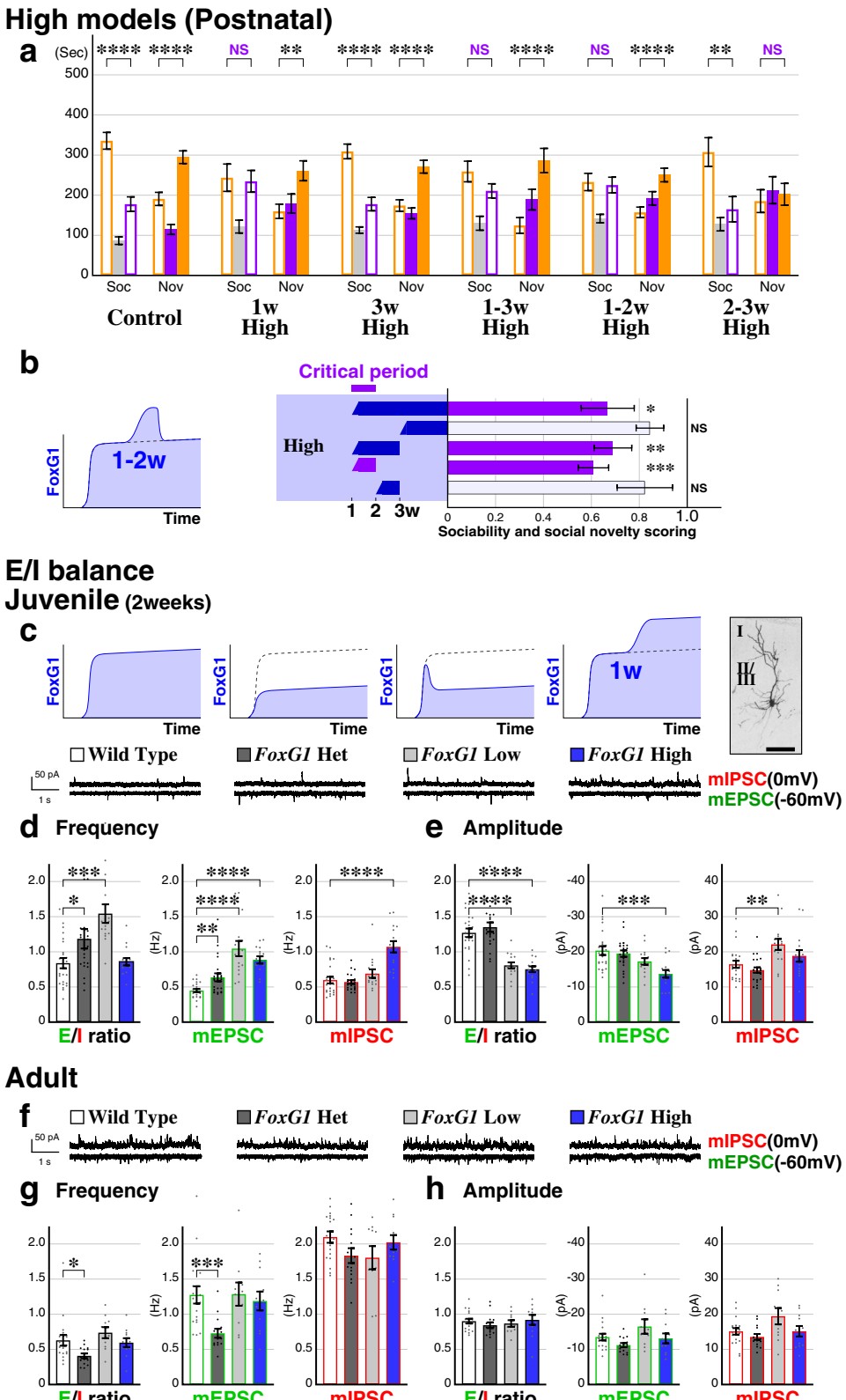

rescue of E/I balance at 2 weeks in Heterozygotes upon GABAergic cell precursor transplantation restores the E/I balance to normalcy in adulthood (Fig. 6e, f).

We next tested if correcting the E/I balance of the mPFC at two weeks would rescue the adult ASD-like social behavioral impairments of the Heterozygous animals. In the three-chamber assay (Fig. 6g and j), GABAergic cell transplantation

into wild type animals did not overtly affect the sociability of the animals at 6 weeks and was comparable to non-transplanted wild type (Fig. 1b) or other control animals (Figs. 2m and 3b). When *FoxG1* Heterozygotes received GABAergic cell precursors into the mPFC at postnatal 1 week (Fig. 6a and h), the animals spent comparable time in the social side of the chamber to the controls during the sociability session (Fig. 6j, NS in orange). During social

**Fig. 4 An early juvenile ASD critical period is revealed by timed *FoxG1* dysregulation and imbalanced excitatory-inhibitory circuits. a, b** In addition to *FoxG1* Glu + GABA High manipulation at 1 week (Fig. 3a, b), *FoxG1* augmentation was carried out at 3 weeks or only during 1–3, 1–2 or 2–3 weeks by replacing doxycycline diet with a regular one (Fig. 3a scheme). While the sociability of 1–3w and 1–2w High animals is altered to a similar extent as the 1w High animals, 3w High model is comparable to control animals. $n = 24, 23, 34, 19$, $p = 3.29 \times 10^{-6}$****, $= 4.28 \times 10^{-5}$****, $= 0.130$, $= 6.87 \times 10^{-5}$****, $= 0.794$, $= 5.10 \times 10^{-5}$****, $= 0.00512$**, $= 0.669$ (from 3w High to right) Sociability scores of the animal models are cross correlated in a manner similar to Figs. 1c and 3c. The ASD critical period is identified as being between 1 and 2 weeks (**b**, left) since *FoxG1* augmentation during this time is sufficient for the animals to develop social behavioral impairments (**b**). $p = 0.0183$*, $0.102$, $0.00591$**, $0.000184$***, $0.206$ (**b**, from top to bottom). **c–h** Miniature EPSCs and IPSCs were recorded from the layer 2/3 pyramidal neurons of the medial prefrontal cortex (prelimbic and infralimbic, **c** right) at the holding potentials of $-60$ mV and $0$ mV, respectively. **c** Examples traces of the Wild Type and *FoxG1* Heterozygous, Low (neuronal Glu + GABA) and High (1 week Glu + GABA) ASD model animals (**c**) at juvenile stages (P12-14). Representative example of a filled layer 2/3 wild type pyramidal cell is shown with a 100 μm scale bar (right). **d, e** The E/I ratio is attenuated in all of the socially impaired *FoxG1* model animals. $n = 23, 22, 15, 15$ for WT, Het, Low and High, $p = 0.0341$*, $0.000111$***, $0.821$, $= 0.00478$**, $7.13 \times 10^{-5}$****, $2.06 \times 10^{-7}$****, $= 0.605, 0.243, 3.70 \times 10^{-5}$**** (**d**, vs WT from left to right for E/I, mEPSC and mIPSC), $p = 0.380$, $4.22 \times 10^{-7}$****, $3.61 \times 10^{-8}$****, $= 0.585, 0.0579, 0.000304$***, $= 0.199, 0.00568$**, $0.230$ (**e**, similar to **d**). **f–h** Data in adult (6–8 weeks) mPFC is shown. $n = 18, 14, 10, 11$ for WT, Het, Low and High, and $p = 0.0148$*, $0.331, 0.743$, $= 0.000578$***, $0.957, 0.640$, $= 0.0582, 0.134, 0.574$ (**g**, similar to **d**) and $= 0.283, 0.572, 0.954$, $= 0.0512, 0.208, 0.830$, $= 0.217, 0.110, 0.985$ (**h**, similar to **d**). Note that no obvious changes in the E/I ratio were found in the adult somatosensory barrel cortices (Supplementary Fig. 6). Data are mean ± SEM, p values are from two-tailed *t*-test.

novelty sessions, cell-transplanted Heterozygotes preferred the stranger side (Fig. 6j, asterisks in orange) as well as spent comparable time in the center chamber to the controls (Fig. 6j, NS in orange). These data suggest that re-balancing the E/I ratio of the mPFC at 2 weeks (Figs. 6c, d) by GABAergic cell precursor transplantation allows *FoxG1* Heterozygous animals to recover from the ASD-like socially impaired state. Furthermore, this is consistent with the notion that *FoxG1* must be simultaneously disrupted in both excitatory and inhibitory populations to precipitate ASD behavioral phenotypes (Fig. 3c), since the transplanted GABAergic cells are wild type for *FoxG1*. Next, in order to test the importance of correcting the circuit E/I ratio at postnatal 2 weeks, we further carried out GABAergic cell transplantation at 3 weeks, instead of 1 week (Fig. 6i). We found that the social behavioral phenotypes of the late-transplanted Heterozygous animals at 6 weeks are distinct from the 1 week-transplanted wild type or Heterozygous animals (Fig. 6g–j) and are comparable to the non-transplanted Heterozygotes (Fig. 1b, scores in Supplementary Fig. 7e). This was consistent even later at 8 weeks (Supplementary Fig. 7f), upon matching the differentiation period of the donor cells to the postnatal 1 week transplantation experiments. These data further support our notion that between postnatal 1–2 weeks of age is the critical time-window to mitigate ASD-related social phenotypes. When working memory was tested, cell transplanted Heterozygotes did not show obvious improvements compared to the non-transplanted Heterozygotes (Fig. 6l, NS), suggesting that GABAergic cell precursor transplantation into the mPFC selectively ameliorates the social-related modality but not the general working memory deficits of the Heterozygous model.

Altogether, we conclude that precisely regulated *FoxG1* expression levels during early juvenile development is crucial, and dysregulation of *FoxG1* expression during this period results in a transient perturbation of E/I balance, resulting in ASD-like gamma EEG patterns as well as social behavioral deficits in adult animals. For the ASD FOXG1 syndrome haploinsufficiency model, in which local inhibitory tone is decreased during the early juvenile critical period, a decrease in GABAergic activity exacerbates the social behavior deficits, whereas re-balancing the E/I ratio by enhancing GABAergic tone ameliorates the social impairments (Fig. 6k, scores compared), thus pinpointing this developmental stage as promising for therapeutic intervention.

## Discussion

The developing brain possesses a remarkable resilience to genetic and/or environmental insults, and has the intrinsic means to adjust its functional properties by scaling excitatory and inhibitory neuronal activity through homeostatic regulation[12]. It has been hypothesized that neurocognitive disorders such as ASD arise when the brain's capacity to buffer the cumulative effect of genetic and/or environmental perturbations on synaptic homeostasis during development is exceeded[61]. Here in this study, we identified that the second postnatal week is the critical period to develop alterations in inhibitory circuits and to precipitate ASD behavioral phenotypes in mature animals. We further demonstrate that behavioral abnormalities in both haploinsufficiency and gene duplication models of FOXG1 syndrome are caused by the simultaneous disruption of *FOXG1* levels in excitatory and inhibitory neuronal circuits. To our surprise, while the E/I balance in cortical circuitry was only transiently disrupted during the critical period in all of our socially impaired *FoxG1* animal models (Heterozygous, Low and High), this perturbation was sufficient to cause behavioral symptoms in adults. We conclude that inhibitory circuit development during the critical period is a key aspect of ASD etiology, and that weakening or augmenting GABAergic tone during this period exacerbates and ameliorates the respective adult ASD social phenotypes.

It was only recently discovered that *FOXG1* duplication and haploinsufficiency both lead to syndromic forms of ASD[24–28,30–32]. Here, we demonstrate that the dose-dependent aspect of FOXG1 syndrome can be modeled in the mouse by implementing our novel genetic strategies. To our surprise, balanced dysregulation of *FoxG1* in both excitatory and inhibitory circuits was necessary to develop ASD-related social behavioral alterations. This is in contrast to what has been shown in conditional mutation studies of syndromic ASD genes such as *Mecp2* and *Ube3a*[8,9]. Our current hypothesis is that proper *FoxG1* levels in both excitatory and inhibitory neurons are necessary to maintain circuit homeostasis. Our cell transplantation study supports this notion since addition of wild type MGE-derived[62] GABAergic precursors into the *FoxG1*-Heterozygous neuronal networks ameliorates the social behavioral deficits (Fig. 6). Importantly, the GABAergic populations affected in our *FoxG1* Low and High manipulations include the local interneurons in the hippocampus, cortex, amygdala and striatum as well as striatal projection neurons, whose malfunction has been also implicated in ASD etiology[34,63]. It will thus be interesting in the future to investigate the contribution of each forebrain GABAergic cell type[64] to the ASD-like social behavioral phenotypes in our *FoxG1* model animals. Considering the central role of *FoxG1* in regulating forebrain development, and its dynamic activity during discrete phases of circuit maturation[20], it is likely that *FoxG1* is an important component of the brain's homeostasis machinery. Thus, perturbations in *FOXG1* expression result in a

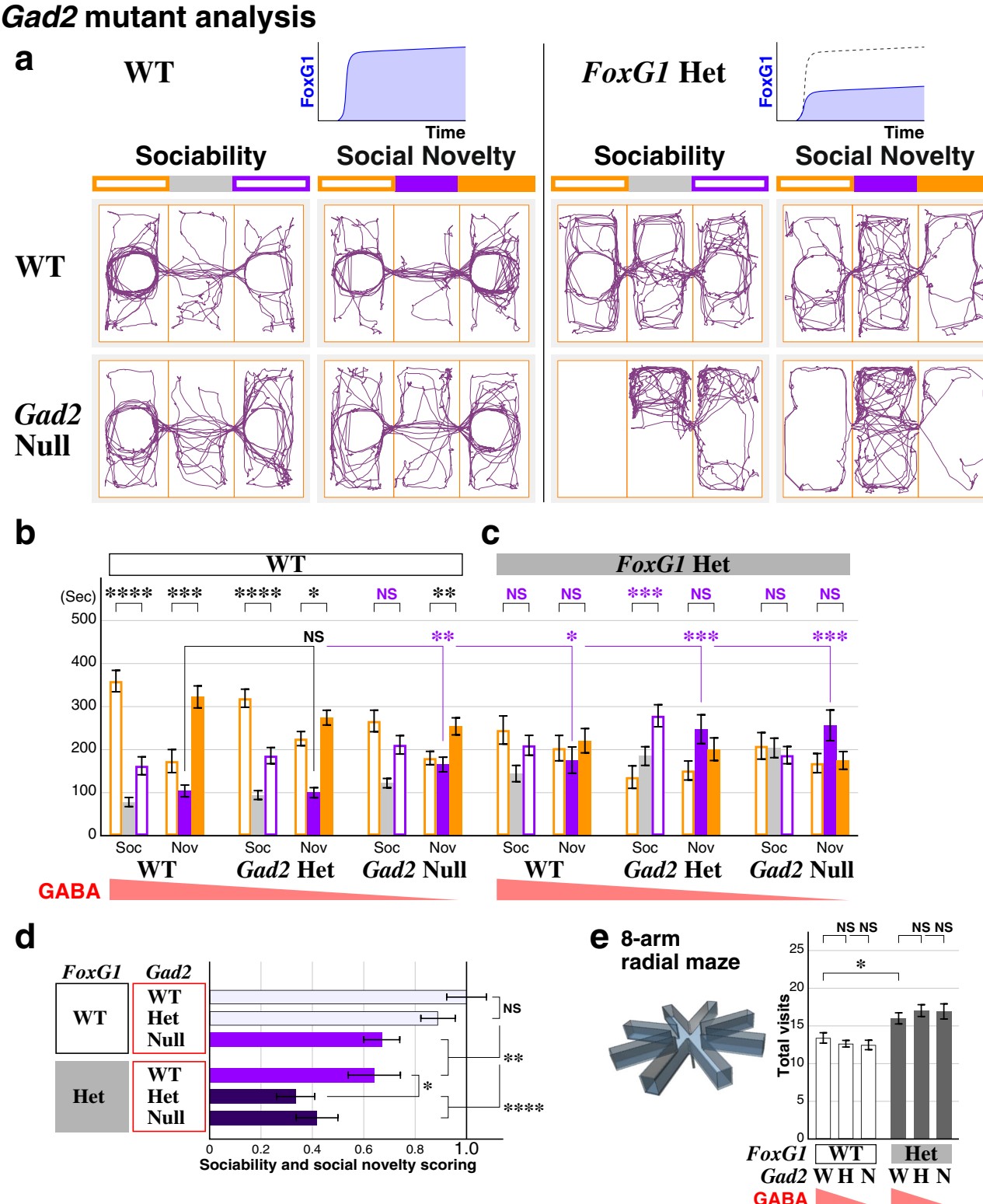

**Gad2 mutant analysis**

failure of homeostasis that may represent a common etiological mechanism for ASD. Consistent with this idea, *FOXG1* may participate in the regulation of mitochondrial function[65], which presents another potential link to ASD etiology[66,67]. In total, we conclude that *FOXG1* likely regulates multiple sets of target genes in a manner analogous to other ASD-related transcriptional regulators such as *MECP2*[68,69] and *CHD8*[70,71].

While *FOXG1* haploinsufficiency and duplication have both been classified as syndromic forms of ASD, there are differences found in the disease phenotypes[24–28,30–32]. Consistent with this, our mouse models for *FoxG1* haploinsufficiency (Het and Low) and augmentation (High) show distinct phenotypic outcomes. While the sociability scores of the Het, Low and High models are comparable (Fig. 3c), High models do not show obvious deficits

**Fig. 5 Decrease in GABAergic tone exacerbates the ASD social phenotypes of the *FoxG1* Heterozygous model.** *Gad2* heterozygous and null littermate animals in the background of wild type and *FoxG1* Heterozygous model were analyzed by three-chamber assay. **a** Example traces of the animals are shown for the four genotypes. **b** In contrast to the *Gad2* Het animals, Null animals showed ASD-like social behavioral alterations by not preferring the social side of the chamber (NS, purple) and staying in the center during the social novelty session (asterisks, purple). **c** *Gad2* Het and Null animals in the *FoxG1* Heterozygous background showed ASD like social behavioral impairments, and during social novelty session, strongly preferred to stay in the center chamber compared to the control animals. n = 22, 28, 21, 25, 29, 20 (**b**, **c**, from left to right), $p = 3.56 \times 10^{-7}$****, 0.000234***, $1.62 \times 10^{-5}$****, 0.0452*, 0.101, 0.00577**, = 0.380, 0.676, 0.000264***, 0.156, 0.562, 0.838 (**b**, **c**, left vs right), = 0.825, 0.00781**, 0.0396*, 0.000326***, 0.000521*** (**b**, **c**, vs WT, center). **d** Social behavior scores of the *Gad2* and *FoxG1* mutant models. While the score of *Gad2* nulls is comparable to the *FoxG1* Heterozygous model (purple bars), *Gad2* Het or Null mutations exacerbates the impaired sociability score of *FoxG1* Heterozygotes (dark bars). $p = 0.283$, 0.00294**, 0.00738**, $1.25 \times 10^{-7}$****, $6.49 \times 10^{-6}$****(vs WT, from top to bottom), = 0.0188*(*Gad2* WT vs Het in *FoxG1* Het background). **e** Working memory is not affected by *Gad2* mutation(s) in both wild type and *FoxG1* Heterozygotes. n = 44, 56, 40, 48, 58, 40 (from left to right), $p = 0.0121$*(WT vs *FoxG1* Het), = 0.352, 0.320, 0.339, 0.460(NS, from left to right). Data are mean ± SEM, $p$ values are from two-tailed $t$-test.

in social novelty assays (Figs. 3b and 4a). In addition, the High model shows a decrease in the E/I amplitude but not the frequency at juvenile stages (Fig. 4d, e) and the values for gamma EEG are not identical to the haploinsufficiency models (Fig. 3l). Thus, while *FOXG1* augmentation has been proposed to be etiologically relevant in idiopathic ASD patients[23], *FOXG1* haploinsufficiency could be a distinct syndrome within the spectrum of ASD. Similar to *FOXG1*, copy number variants in *MECP2* with an extra copy (Duplication syndrome) or deletion (Rett) have both been observed in syndromic ASD patients[1,72]. While Mecp2 and FoxG1 have been shown to interact directly in neuroprotection studies[73], the clinical presentations of *FOXG1* and *MECP2* related syndromic ASDs are distinct[74]. Consistent with this idea, in contrast to the *Mecp2* mutants, *FoxG1* adult mouse models do not exhibit increases in cortical E/I activity.

It has long been suggested that GABAergic neuron dysfunction contributes to the pathophysiology of ASD[2,3,5,6]. Although a central role for the critical developmental period of inhibitory circuits in ASD etiology has not been proposed, recently, by using a juvenile social isolation paradigm, the proper maturation of *Pvalb*-positive interneurons in the mPFC has been shown to be important for animal social behavior[75]. While this study highlights the social experience- (3 to 5 weeks) and interneuron-dependent (*Pvalb* subtype) critical period of mPFC for social behavior, our study defines an innate developmental critical period for mPFC social circuits and ASD. Interestingly, by analyzing intrinsic electrophysiological properties[76], we found clear alterations in putative *Pvalb* subtype (Fast-spiking) interneurons of the *FoxG1* Heterozygous model already at two weeks, in contrast to pyramidal cells that were found to be abnormal at later timepoints (Supplementary Fig. 8).

Cortical activity during early postnatal development has been shown to facilitate cortico-striatal coupling, and decreasing the cortical hyperactivity observed in the *Shank3B* mutant ASD model mouse during this period was able to rescue the striatal phenotypes[63]. Moreover, in rodent ASD models, the connectivity between the mPFC and areas such as the striatum, basolateral amygdala and ventral tegmental area has been proposed to be disrupted[77]. By analyzing *Gad2* mutant animals, we provide direct evidence that a decrease in overall GABAergic synaptic tone leads to ASD-like social behavioral impairments, to a comparable extent as observed in the FOXG1 syndrome model. In addition, our *Gad2* mutant studies revealed that a reduction in GABAergic tone further exacerbates the ASD social phenotypes of the *FoxG1* model. This strongly supports the notion that transient circuit E/I imbalance during the critical period does not simply reflect homeostatic compensation per se, but rather is a causative driver of ASD phenotypes in adult FOXG1 syndrome model animals. Thus, it will be interesting to investigate the E/I balance in juvenile animals of other established ASD models[12]. Consistent with mPFC circuits playing central roles in regulating

social behavior, social impairments have been rescued in several contexts by directly manipulating mPFC activity while animals are executing behavioral tests[4,51,52,75]. Here in this study, we were able to developmentally ameliorate the sociability deficits in our ASD model animals. Thus, our findings on the putative ASD critical period pinpoints a developmental stage in GABAergic circuits that offers a possible route for therapeutic intervention.

## Methods

***FoxG1* perturbation model animals.** All animal handling and experiments were performed in accordance with protocols approved by the respective Institutional Animal Care and Use Committees of the NYU School of Medicine, Tokyo Metropolitan Institute of Medical Science and Tokyo Women's Medical University. Animal cages are maintained at 22 °C ± 1 °C, 50 ± 15% humidity with a 12 h light/dark cycle. To generate the *FoxG1* Heterozygous model, we combined a *FoxG1 LacZ* knock-in null allele[13] with the wild type *FoxG1* allele. *FoxG1* Low models were generated by crossing male *Nex-Cre*[43]; *Dlx-Cre* animals[44] with female *FoxG1-C/C*; *R26-CAG-FRTed stop-EGFP* homozygous animals[59] (RCE:FRT, Jackson Laboratories stock #10812). Upon Cre-mediated recombination, the conditional floxed-*FoxG1* allele is converted into a *Flpe* knock-in allele[20], and thus the recombination efficiency in each animal can be evaluated by EGFP expression from the Flp-dependent EGFP reporter RCE:FRT. *FoxG1* High models were generated by crossing male *Nex-Cre*; *Dlx-Cre* animals to female *R26-stop-tTA* homozygous[78] (Jackson Laboratories stock #008600); *TRE-FoxG1* homozygous animals[79]. In order to delay *FoxG1* augmentation until after postnatal 1 week, a doxycycline-containing diet (200 PPM, #1810413 Test Diet) was provided to the pregnant females from the date of the observed plug until postnatal day 7. 3 week High and 1–3 week High models were generated by replacing the doxycycline diet with a regular one at P21 and during P7 to P21, respectively. Other variations of 1–2 and 2–3 week High animals were generated in a similar manner. The *TRE-FoxG1* allele contains an *IRES-LacZ* cassette downstream of the *FoxG1* coding sequence and thus, the corresponding *FoxG1* GOF population can be visualized based on X-gal staining or by using anti-beta-galactosidase immunohistochemistry. In order to carry out littermate studies for *FoxG1;Gad2* compound mutants, double-heterozygous male *FoxG1-LacZ*; *Gad2* animals were crossed to female *Gad2* heterozygotes[53,54].

**Tissue preparation and immunohistochemistry.** Embryos were dissected out, and transcardiac perfusion was performed with cold 4% formaldehyde/PBS solution (w/v). Brain dissection was carried out in PBS in a petri dish, and brains were subsequently placed into 4% formaldehyde/PBS solution on ice. The P7 to P21 as well as adult brains were also perfused with the same fixative. Brains were postfixed according to developmental stages (E14.5: 20 min, E16.5: 40 min, E18.5 onwards: 60 min) and, following a brief rinse in PBS, were placed into 25% sucrose/PBS (w/v) solution for cryoprotection until they sunk to the bottom. Subsequently, brains were mounted in OCT compound (Sakura Finetek) and stored at −80 °C. Cryosections were prepared at 12 μm thickness and collected on glass slides (Fisher or Matsunami), and, after over 1 h of drying, the sections were stored at −80 °C. Immunohistochemistry was performed similarly to previously described[20,59,60]. All reactions were carried out in PBS containing 1.5% normal donkey serum (Jackson Immunoresearch) (v/v) and 0.1% Triton X-100 (v/v) (DST solution), which we generally keep at 4 °C up to two weeks. By using a liquid blocker pen (#Z377821, Sigma), two lines are placed in between the section nearest to the frost of the slide and the frost to restrict the movement of liquid on the slide glass. Sections were rinsed 3 times in PBS to remove residual OCT, followed by non-specific antibody blocking, which was carried out in DST solution at room temperature for 30–60 min. Primary antibody incubation was subsequently performed overnight at 4 °C. Primary antibody was included in the DST solution at the following concentrations: Rabbit anti-GFP (1:2000; Molecular Probes, #A11122), Rat anti-GFP #GF090R (1:2000; Nacalai Tesque, #04404-84), Rabbit anti-RFP Living Colors DsRed Polyclonal Antibody (1:2000; Takara Bio Clontech, #632496), Rat anti-RFP

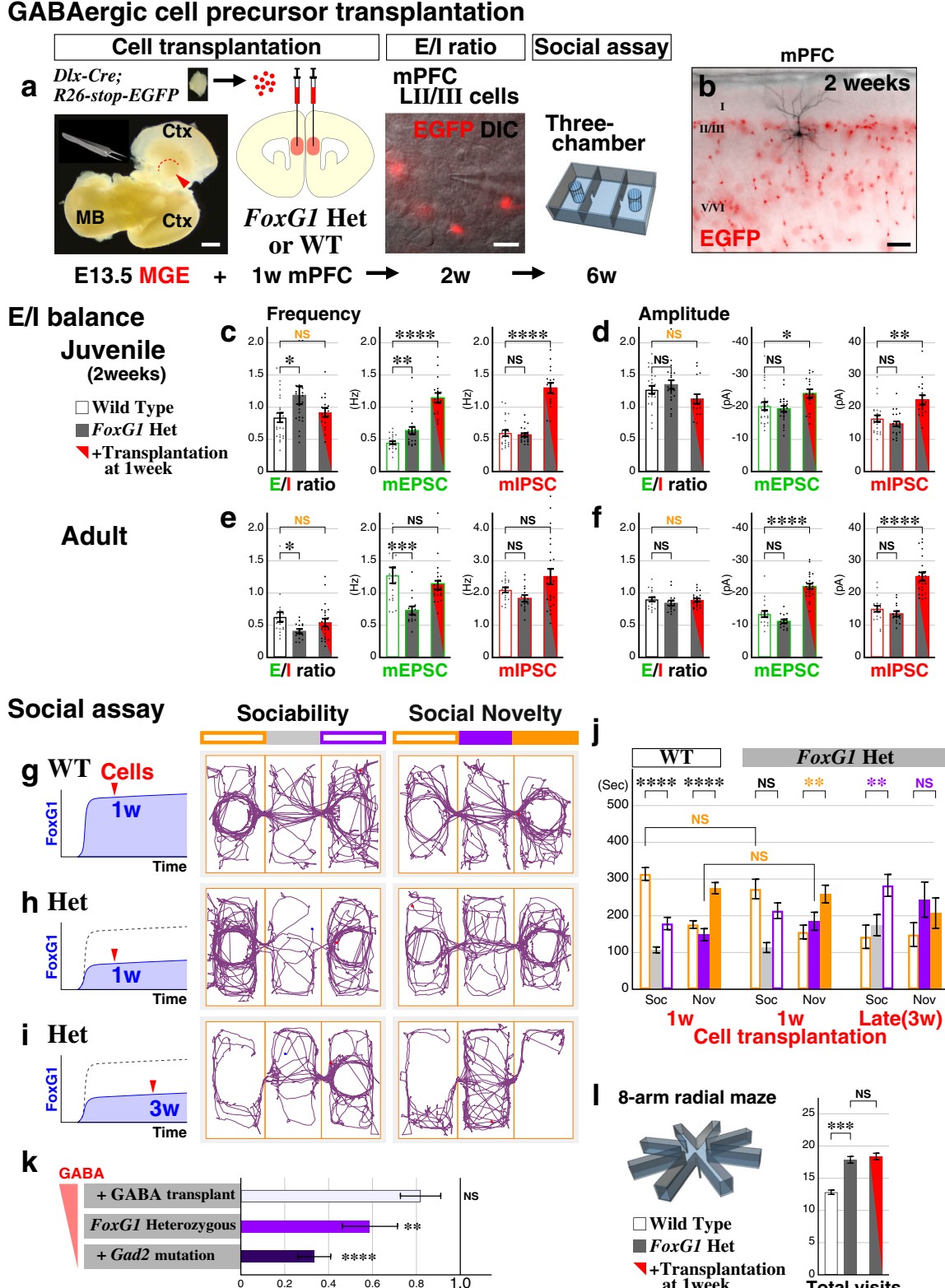

**GABAergic cell precursor transplantation**

#5F8 (1: 2000; Allele Biotechnology, #ACT-CM-MRRFP10). Secondary antibodies conjugated with Alexa Fluor dyes 488 or 594 for anti-Rabbit Alexa 488 (Invitrogen, #A-21206), anti-Rat Alexa 488 (Invitrogen, #A-21208), anti-Rabbit Alexa 594 (Invitrogen, #A-21207), anti-Rat Alexa 594 (Invitrogen, #A-21209) raised in donkey and cross-adsorbed against other species were used in 1:2000 dilution for fluorescent immunohistochemistry. In order to visualize the cell nuclei and laminar structure of the developing neocortex, DAPI (Sigma) (1 ng/μl in PBS and filtrated) was added on slides for 20–30 min following the immunohistochemistry staining

procedure described above. Slides were rinsed a couple of times in PBS and then coverslipped using Fluoromount G (Southern Biotech). Fluorescent images were captured with a Zeiss AxioScope.A1 (Carl Zeiss) by using a cooled-CCD camera (Princeton Scientific Instruments, NJ) with Metamorph software (Universal Imaging, Downingtown, Pennsylvania) or by using a QSI RS 6.1s cooled-CCD camera (2758×2208 pixels, QSI) with micro-manager (open source software). Fluorescent in situ hybridization was carried out by using a *Gad1* anti-sense probe and by visualizing signals with Alexa Fluor 594 Tyramide Reagent (#B40957, Thermo

**Fig. 6 Normalization of the mPFC E/I balance by incrementing GABAergic tone during the critical period ameliorates social phenotypes in the *FoxG1* ASD model. a** Donor GABAergic cell precursors were dissected out from the MGE (medial ganglionic eminence, arrowhead) of E13.5 embryos and dissociated. Subsequently, donor cells were bilaterally transplanted into the mPFC (medial prefrontal cortex) of 1 week old *FoxG1* Heterozygous or Wild Type animals. The E/I balance of layer 2/3 pyramidal cells was analyzed at 2 weeks (DIC image) and the social behavioral assay was carried out at 6 weeks. **b** A representative example of cell-transplanted mPFC slices after the E/I analysis at 2 weeks. A biocytin-filled pyramidal cell and EGFP-labeled GABAergic cell transplants are shown. Control wild type and the Heterozygous animals maintained a similar density of transplanted cells within the adult mPFC (Supplementary Fig. 7b–d). **c–f** E/I balance data for frequency and peak amplitude values at 2 weeks (**c**, **d**, $n = 16$) and in adults (**e**, **f**, $n = 22$). Cell-transplanted Heterozygotes showed comparable E/I ratios to the control Wild Type animals (NS in orange) for both frequency ($p = 0.424$) and amplitude ($p = 0.148$) at 2 weeks and in adulthood ($p = 0.379$ and $0.828$). $p = 8.04 \times 10^{-8****}$, $7.47 \times 10^{-8****}$ (**c**, mEPSC and mIPSC), $= 0.0438*$, $0.00294**$(**d**), $= 0.291$, $0.115$ (**e**), $= 8.69 \times 10^{-8****}$, $2.95 \times 10^{-7****}$ (**f**). **g–j** Upon receiving GABAergic cell precursors at 1 week (**g**, **h**, left scheme), social behaviors of Wild Type ($n = 34$) and Hets ($n = 29$) were analyzed by three-chamber assay (**g**, **h**). In addition, in order to test the importance of rebalancing E/I ratio of the Hets during the critical period (1–2 weeks), cell transplantation was carried out at postnatal 3 weeks (**i**, $n = 21$). Time spent in the social side of the chamber during sociability sessions (NS in orange, $p = 0.202$) and center chamber during social novelty sessions (NS in orange, $p = 0.222$) were comparable between the 1 week-transplanted Wild Type and Heterozygotes (**j**). $p = 4.48 \times 10^{-7****}$, $1.32 \times 10^{-6****}$, $0.0877$, $0.00126**$, $0.00264**$, $0.275$ (**j**, left vs right, from left to right). **k** The impaired sociability score in the Heterozygotes is ameliorated ($p = 0.122$) and exacerbated in the GABAergic cell precursor transplantation and *Gad2* mutation models, respectively. **l** In contrast to social behavior (**j**, **k**), working memory deficits of the Heterozygotes were not ameliorated upon GABAergic cell transplantation into the mPFC ($n = 54$, $p = 0.698$). Data are mean ± SEM, $p$ values are from two-tailed $t$-test. Scale bars: 1 mm (**a**, left), 20 μm (**a**, right), 100 μm (**b**).

Fisher Scientific) following a similar protocol similarly to previously described[80]. In order to facilitate the visualization of gene expression, acquired images were inverted in Photoshop (Adobe) and subsequently combined to generate the figures. For example, tdTomato fluorescent images were placed into both the green and blue channels of the RGB format file and then all color channels were simultaneously inverted to generate a figure with red signals in the white background. For a multicolor presentation, layers including different color signals were assembled by using the function of multiply layers.

**Western blotting of mPFC tissue.** Brain slices were prepared in a similar manner to the in vitro electrophysiological recordings (300 μm thickness) or by generating a coronal slice with a cutting blade (typically ~400 μm). In the petri dish, mPFC was bilaterally dissected out from the slice with forceps, placed into a 1.5 ml tube and immediately stored in −80 °C freezer. After obtaining tissue samples from the Het, Low and High models at various postnatal stages, and in addition, E16 embryonic cortical Het and Null tissue as positive controls, protein lysate was prepared. For each sample, 200 μl of SDS lysis buffer (1% SDS, 10 mM Tris pH7.5, 5 mM EDTA, 1tablet/10 ml Protease inhibitor cocktail (Roche: 11-836-153-001), 1tablet/10 ml PhosSTOP (Roche: 04-906-837-001)) was added and homogenized with pipetting. After vortex mixing the samples, tubes were centrifuged for 15 min with $20,630 \times g$/15,000 rpm and subsequently, the supernatant protein concentration was analyzed by using the BCA Protein Assay Kit (Thermo Scientific #23227). Total 4 μg of protein lysate from each sample was applied to the well of Mini-Protean TGX 4–15% Gel (BIORAD #456-1086) for western-blotting analysis. We initially compared three FoxG1 antibodies of Rabbit anti-FoxG1 (Neuracell, #NC-FAB), Rabbit anti-FoxG1 (Abcam, #ab18259) and Rabbit anti-FoxG1 monoclonal # EPR18987 (Abcam, #ab196868) in 1:10,000 concentration for E16 wild type and *FoxG1* null cortical tissues. We decided to use the rabbit monoclonal # EPR18987 (1:10,000 concentration) because we detected almost no background bands in Null tissues, as well as the positive band was clear (Supplementary Fig. 5a). Gel blotting was performed with a regular procedure by using a 0.45 μm Nitrocellulose membrane (Biorad #1620116) or Immobilon PVDF membrane (Millipore#IPVH07850) and after using 5% ECL blocking reagent (GE Healthcare # RPN2125V) in TBST, antibody reactions were carried out for rabbit mono FoxG1 or mouse monoclonal βactin antibody #AC-15 (1:50,000; SIGMA Aldrich #A5441-2ml) or mouse monoclonal Tuj1 antibody #TU-20 (1:5000; Millipore # MAB1637). After 1 h of secondary reaction with 1:5000 HRP-conjugated anti Rabbit or Mouse antibodies (Jackson ImmunoResearch #115-035-003 or 111-035-003) and subsequent washing of the membrane, signals were recorded by Clarity Western ECL Substrate (Biorad #1705061) and Image Quant LAS 4000 (GE Healthcare). When required, antibody stripping from the membrane was performed by 30 min incubation at 55 °C with a stripping solution containing 2% SDS, 100 mM mercaptoethanol, and 62.5 mM Tris-HCl (pH 6.8). Obtained signals were analyzed by using the Image J software.

**GABAergic cell precursor transplantation.** In order to prepare donor EGFP-labeled GABAergic cell precursors, we crossed males double positive for *Dlx5a-Cre*[44] (Jackson Laboratories stock #008199) and *R26-CAG-loxPstop-EGFP* reporter homozygous[59,81] (*RCE:loxP*; Jackson Laboratories stock #10701) to wild type female animals. At E13.5, embryos were dissected out and the ones harboring green brains were confirmed by using a fluorescent dissection microscope (Olympus, SZX 12). From each embryo, two MGEs (medial ganglionic eminence)[62] from both hemispheres were collected. We typically used total 8 embryos for donor but in case we do not have 8 EGFP-positives, we combined EGFP-negative embryos as

well. For cell dissociation, brains were dissected out in HBSS solution (Gibco #14175-095) one by one, each pair of MGEs were dissected out and stored in a 1.5 ml flat-bottom micro tube (Watson #131-815CS) with 500 μl of HBSS solution and kept on ice until the dissection was complete for maximum 8 embryos. Later, tissues were rinsed twice with HBSS and 100 μl of DNAse I (20,000 ku/ml in HBSS, Worthington) and 400 μl of Papain (20 u/ml in HBSS, Worthington #LK003176) solution were immediately added. By gentle pipetting (typically around 5 times) with a 200 μl filter tip pipette, tissues were broken up into small pieces. After a 20 min of incubation at 37 °C with shaking, 750 μl of 1% (v/v) heat-inactivated horse serum (Gibco #26050-070) in ice-cold HBSS and 75 μl of DNAse I (2000u/ml in HBSS) were added. After gently inverting the tube a couple of times, the tube was centrifuged for 5 min at 200 g/1480 rpm at 4 °C, the supernatant was discarded, 500 μl of ice-cold DMEM/GlutaMax (containing 5 μl of DNAse I) was added, and the cell pellet was resuspended by gentle pipetting using a 200 μl filter tip pipette. Cells are then collected into a 1.5 ml flat-bottom tube through 40 μm mini cell-strainer II (Hi-tech-inc# HT-AMS-04002, blue). The cell-strainer is rinsed with another 500 μl of ice-cold DMEM/GlutaMax containing 5 μl of DNAse I to make total 1 ml cell solution. Cell numbers were counted on OneCell counter (Fine Plus International Ltd). MGEs from 8 embryos typically ended up with 6000–8000 cells/ μl in 1 ml cell solution. 1 ml of cell solution was centrifuged for 5 min at 4 °C, 200 g/1480 rpm and supernatant was removed to leave 15-20 μl with the cell pellet. After an addition of 0.5 μl of DNAse I (2000 u/ml in HBSS), the tube was vortex mixed and kept on ice for cell transplantation surgery.

In the afternoon, P7 pup was placed in a bag and buried under crushed ice for 15 min to have sufficient anesthetization. For 3 week old experiments, animals were first anesthetized with 2% isoflurane and then intraperitoneal injection of ketamine (100 mg/kg) and xylazine (16 mg/kg) mixture was carried out. 3 μl of cell solution was placed on a parafilm (typically for 3–4 pups) and was adsorbed into a 36 g beveled NanoFil injection needle (World Precision Instruments, #NF35BV-2) at 300 nl/min ratio. Surgical tape-protected ear-bars are placed bilaterally to fix the pup head. The skin of the head above the olfactory bulbs was cut in the center for about 3–5 mm to visualize the skull. 1.7 mm posterior and 0.2 mm lateral from the blood vessel crossing, which is located above the border of nasal and frontal skull was aimed. Prior to needle injection, cells are released from the needle to confirm the clogging. The needle tip is placed 1.5 mm deep from the surface, and 200 nl of cells are injected at 200 nl/min speed. Upon completing the cell injection, the needle was kept still for another minute and then removed from the skull slowly. Upon completing the cell injection into the right hemisphere, pup status was confirmed, and if necessary, a small ice pack was placed below the belly of the pup before proceeding to the left hemisphere injection. When done with the injections into both hemispheres, the head was thoroughly wiped, the skins were put together with instant glue and the pup was placed on the warm pad for a while before sending back to the littermate cohort. The entire surgery takes approximately 25 min for one pup.

**Behavioral assays.** The date of the observed plug was designated as E0 and pups were usually delivered on E19. At postnatal day 21, the whole litter was weaned and placed into a larger cage (W270 × L440 × H187 mm, KN-601, Natsume Seisakusyo) from the original cage (W220 × L320 × H135 mm). Genotyping of the animals by tail PCR was carried out by postnatal 2 weeks. Upon completing battery of behavioral analyses, we again carried out tail PCR to assure the genotypes of the animals. Open field and elevated plus maze behavioral assays were carried out on cohorts of animals at 5 weeks of age. The following week, the three-chamber assay

and 8-arm radial maze working memory task were performed. Mice of both genders were included in our analysis. All of the assays were carried out in a sound-proof room (S-1220 soft DX, STAR LITE) and mouse behavior in each specific arena was recorded by a CCD camera (Chameleon3 1.3 MP Mono USB3 Vision, CM3-U3-13S2M-CS, FLIR) at 15 frames/sec and saved as M-JPEG file with 75% compression by using FlyCapture2 camera software 2.9.3.11 (Photonics). Prior to initiating the recording, the information of the test animal, such as date of filming, animal number, and presence or absence of a social animal, were written on a small white board (22 × 33 cm) and placed on the behavior testing set. After video recording had been initiated, the white board was removed and the test animal was placed in the arena or the starting dome was removed to release the test animal. Once testing had begun, the experimenter would quickly leave the sound-proof chamber and quietly close the door in order to provide a non-disturbing environment to the test animal during the assay. The experimenter was always blind to the genotype of the test animals. Video files were later analyzed by using the ANY-maze video tracking software 5.1 (Stoelting, USA), with the video analysis typically initiated after the door had been closed. All behavioral analysis data are shown as mean ± SEM and $p$ values are from two-tailed $t$-test.

**Open field assay.** The open field apparatus consists of a gray acrylic-modified polyvinyl-chloride floor with a 50 × 50 cm square field and gray vinyl chloride 40 cm height walls (OF-36MSQ, Muromachi). Animals were placed in the center of the arena and the recording was performed for 10 min and 30 s. Data were analyzed over the 10 min session for the distance traveled and time spent in the center (30 × 30 cm square center) and margin of the chamber.

**Elevated plus maze.** The apparatus material is similar to that of the open field chamber, and the length of each of the open and closed arms is 30 cm, with a corridor width of 6 cm (EPM-04M, Muromachi). The walls of the closed arm are 15 cm in height. At the beginning of the assay, animals are placed on one end of the open arms and allowed to explore freely during the 10 min and 30 s recording session. Distance and time each animal spent on the open and closed arms were analyzed for the first and second 5 min of the assay.

**Three-chamber social interaction assay.** The apparatus consists of a gray acrylic-modified polyvinyl-chloride floor and three chambers (each 20 × 40 cm) connected by 5 cm width × 7 cm height windows in transparent acryl 22 cm height walls (SC-03M, Muromachi). Wired cages are 18.5 cm height with 9 cm diameter circle bottom and top, both of which are connected by 3 mm diameter wires (total 16) placed in a circle with 7 mm distance between wires. Out of the four wired cages, two were used to place the stranger mouse but the other two did not receive animals and were used for the Habituation (1st 10 min) and non-social side of the Sociability session (2nd 10 min). The chambers and wired cages were cleaned and dried with paper towels between each test animal trial. The test mouse is first placed in a start dome (20 cm diameter circle tube, transparent acryl) in the middle chamber. Upon removal of the start dome, the experimenter quickly left the sound-proof room and closed the door quietly. In the first session (Habituation), the animal was allowed to freely move inside the three-chambers with two empty wired cages placed in the centers of the lateral chambers. After 10 min and 30 s video recording was obtained, the test animal was allowed to move into the middle chamber and was trapped in the start dome. At the beginning of second session (Sociability), one of the empty wired cage was replaced with another wired cage holding an age- and gender-matched stranger mouse. After 10 min and 30 s of video recording, the test animal was again trapped inside the starting dome in the middle chamber. Then, the remaining empty wired cage was replaced with a stranger animal-containing wired cage for the final session (Social Novelty). After the video recording was complete, the test animal was returned to its littermate home cage. For each 10 minutes of assay, time spent in each of the three chambers was measured and analyzed.

**8-arm radial maze working memory task.** The maze is made from a material which is similar to the open field, and each arm is 35 cm length with 6 cm wall heights (RM-08M, Muromachi). The center part of the arms (10 cm) have higher walls (12 cm) and the distance between the opposing two arms spanning the center area is 20 cm. At the end of each arm, there is a circle hole (2 cm diameter, 1 cm depth) with a small dip (0.5 cm diameter, reverse cone shape) in the center. Prior to initiating the task, 50 µl of water droplets are carefully placed into all of the eight dips by using a pipette. The test animal is placed in a start dome (20 cm diameter circle tube, transparent acryl), which is placed in the center of the arena. The assay starts by removing the start dome and the experimenter leaving the sound-proof chamber and closing the door. In terms of the test model animals, water bottles were removed from the cage in the morning of one day prior to the first experiment. On the first day, assays are carried out for two rounds (R1 and R2) typically in the morning and the afternoon. On the second day, another two rounds of experiments (R3 and R4) are carried out. In all of the tests, video recording was carried out for 5 min and 30 s. With the ANY-maze software, the analysis of video file is initiated when the test animal reaches the first end of the arm by its body

trace dot crossing a line located 10 cm away from the end of arm. When the test animal has entered the eighth end of the arm, the trial is terminated; otherwise it ends after 5 min. The duration to complete the task, total numbers of the end of the arms visited and the mean speed was measured for each animal in the R3 and R4 trials for analysis. Note that when a test animal does not complete the task within 5 min, the total numbers of visits, as well as duration, are both underestimated.

**Electroencephalogram recording**

*EEG and EMG recordings of freely-moving animals.* Stereotaxic surgery was performed under anesthesia with a ketamine-xylazine mixture (100 and 10 mg/kg, respectively, i.p.). Electrodes for EEG were implanted on the skull over the left and right prefrontal cortex (+3.0 mm anteroposterior, +1.5 mm mediolateral from bregma; +1.1 mm anteroposterior, +1.5 mm mediolateral from bregma) according to the mouse brain atlas[82]. Electrodes for EMG were inserted into the neck muscles. Continuous EEG and EMG recordings were performed through a slip ring (SPM-12, Hikari Denshi Kogyo) designed so that the movement of mouse was unrestricted. EEG and EMG signals were amplified (BA1008, Miyuki Giken), low-pass-filtered at 100 Hz, digitized at a sampling rate of 400 Hz, and recorded using LabChart v7.3.8 software (AD instruments). For the vigilance state determination, EEG/EMG recordings were scored by visual inspection as wakefulness, NREM sleep, or REM sleep in 8-sec epochs following the criteria[83]. The same individual, blinded to experimental conditions, scored all EEG/EMG recordings. Spectral analysis of the EEG was performed by fast Fourier transform (SleepSign software version 3.0, Kissei Comtec). The EEG signal were filtered by 0.5 Hz high-pass filter and 50 Hz notch filter. This analysis yielded a power spectral profile over a 0.5–80 Hz window with a 0.2 Hz resolution divided into delta (0.75–4 Hz), theta (4–12 Hz), alpha/beta (12–30 Hz), low-gamma (30–45 Hz), and high-gamma (55–80 Hz) waves. Whereas the recordings were obtained for more than 24 hr, the EEG data in wakefulness state early in the dark period (from 22:00 to 0:00) was used for the spectral analyses. The absolute power was divided by the total power (0.5–80 Hz) in 24 hr. The mean normalized power in each frequency band was then calculated for each subject.

*Sensory-evoked EEG recordings in awake head-fixed animals.* In order to prepare for EEG recording, a custom-made head plate was implanted onto the skull of 7–8 week-old adult *FoxG1* model animals using acrylic dental resin (Super-Bond, Sun Medical, Shiga, Japan). Prior to this surgery, mice were anesthetized with an intraperitoneal injection of ketamine (100 mg/kg) and xylazine (16 mg/kg) mixture and held in a stereotaxic apparatus. The head plate-implanted animals were placed back in the home cage and allowed to recover from the surgery for 2 or 3 days. For EEG recordings, the animals were first anesthetized with 2% isoflurane to insert screws. A reference screw was inserted into the skull on top of the right cerebellum and recordings were from a screw on top of the right cortex (1 mm anterior and 0.5 mm lateral to bregma). EEG recordings were obtained from head-fixed mice during the awake state. The EEG signal was amplified and digitized at 4 kH by Plexon OmniPlexD system Ver. 1.12.0.0 (Plexon Inc., Dallas, TX, USA). EEG data were analyzed using a custom-written MATLAB program (MathWorks, Natick, MA, USA). Whiskers were stimulated by using a piezoelectric device controlled by a custom-written MATLAB program. The codes are available from the corresponding author upon reasonable request. All whiskers were deflected toward forward and backward directions for 30 ms with a 170 ms steady state interval after each deflection. This stimulation protocol was carried out four times in one session, which was repeated 20 times at 3.6 s intervals. Recordings with and without stimulation were carried out for 1.6 s after and before the first whisker deflection, respectively.

**In vitro electrophysiological recordings.** Juvenile (P12-14) and adult (6–8 weeks) male or female mice were deeply anesthetized with pentobarbital (100 mg/kg, intraperitoneal) and the brains were quickly removed and submerged in an ice-cold cutting solution (234 mM sucrose, 2.5 mM KCl, 10 mM $MgCl_2$, 0.5 mM $CaCl_2$, 25 mM $NaHCO_3$, 1.25 mM $NaH_2PO_4$, 0.5 mM myo-inositol, and 11 mM glucose). Coronal brain slices were cut using a vibratome (300 µm, VT-1200S, Leica) and immersed in a physiological solution (125 mM NaCl, 2.5 mM KCl, 1 mM $MgCl_2$, 2 mM $CaCl_2$, 26 mM $NaHCO_3$, 1.25 mM $NaH_2PO_4$, 20 mM glucose, 5 mM Na-ascorbate, 3 mM Na-pyruvate, and 2 mM thiourea) that was continuously bubbled with a mixture of 95% $O_2$ and 5% $CO_2$. Na-ascorbate, Na-pyruvate, and thiourea were omitted during recordings. Whole-cell patch clamping was performed at 31–32 °C. Voltage-clamp recordings were made from visually identified layer 2/3 pyramidal neurons in the prelimbic cortex (PFC, prefrontal cortex) or barrel fields of the primary somatosensory cortex (S1).

The pipette solution for current-clamp recordings consisted of 130 mM potassium gluconate, 0.2 mM EGTA, 2 mM $MgCl_2$, 2 mM $Na_2ATP$, 0.3 mM NaGTP, and 10 mM HEPES. The pipette solution for voltage-clamp recording consisted of 108 mM cesium gluconate, 0.4 mM EGTA, 2.8 mM NaCl, 5 mM TEA-Cl, 4 mM MgATP, 0.3 mM NaGTP, 10 mM $Na_2$-phosphocreatine and 20 mM HEPES. In some recordings, 0.5% biocytin was included in the pipette solution to visualize neuron morphology. The pH of the solution was adjusted to 7.2 and the

osmolarity was 290 mOsm. The membrane potentials were not corrected for liquid junction potentials. The series resistance of the recording cells was <25 MΩ. For current-clamp recordings, the resting membrane potentials and firing responses to depolarizing current pulses (1-s duration, 0–600 pA with 50 pA increments from the resting membrane potential) were recorded within 5 min from whole-cell break-in. The input resistance and time constant were determined from the transient voltage response to a hyperpolarizing current injection (−50 pA). The voltage sag amplitude was determined by subtracting the steady state amplitude from the maximum amplitude in response to a hyperpolarizing current pulse (−200 pA, 1-s duration). For voltage-clamp recordings, miniature excitatory postsynaptic currents (mEPSCs) and miniature inhibitory postsynaptic currents (mIPSCs) were recorded as inward currents at −60 mV and outward currents at 0 mV, respectively, from the same neuron in the presence of physiological solution containing 0.5 μM tetrodotoxin (Alomone Labs, Jerusalem, Israel). After recordings, slices containing biocytin-loaded cells were fixed overnight at 4 °C in a fixative (4% paraformaldehyde and 0.2% picric acid in 0.1 M PB) and incubated with Alexa Fluor 488-conjugated streptavidin (1:500, Thermo Fisher Scientific, Waltham, MA, USA).

Recordings were amplified with a Multiclamp 700A amplifier (Molecular Devices, LLC, Sunnyvale, CA, USA), digitized at 10 kHz using a Digidata 1322A apparatus (Molecular Devices, LLC), and collected with pClamp 8.2 software (Molecular Devices, LLC). mEPSC/mIPSC events were detected with a scaled-template algorithm and analyzed using IGOR Pro 6.36 software (WaveMetrics, Inc., Lake Oswego, OR, USA) and Clampfit 8.2 software (Molecular Devices, LLC). The amplitude and the frequency of mEPSC/mIPSC events were determined for each neuron over a 5-min period of recording. Statistical analyses were performed as follows using Graphpad Prism 7.05 software (GraphPad Software, San Diego, CA, USA): two-way ANOVA and post-hoc Tukey's test were used for comparisons of current-frequency relationships; Welch's $t$-test was used for two group comparisons; and the chi-square test was used for comparison of firing type composition.

**Reporting summary**. Further information on research design is available in the Nature Research Reporting Summary linked to this article.

## Data availability

All data supporting the findings of this study are provided within the paper and its Supplementary information. A Source data file is provided with this paper. We confirm that all unique materials used here are readily available from the authors (*FoxG1-LacZ, floxed-FoxG1, TRE-FoxG1, Nex-Cre, Gad2 null*) or from Jackson labs (*Dlx5a-Cre, RCE: FRT, RCE:loxP, R26-stop-tTA, Ai9*). Further information on research design is available in the Nature Research Reporting Summary linked to this article. Any data are available from the authors upon request.

## Code availability

A custom code was used in the EEG analysis of head-fixed animals (Supplementary Fig. 4). All scripts used to analyze or display the data are available upon request.

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

## Acknowledgements

We thank the following Drs. for kindly sharing their reagents with us: Eseng Lai (*FoxG1-LacZ* knock-in mutant), Carina Hanashima (*TRE-FoxG1* transgenic), Marc Ekker (*Dlx5a-Cre* transgenic), and Klaus-Armin Nave (*Nex-Cre* knock-in). We also wish to thank Drs. Kazutaka Ikeda and Hiroko Kotajima for helpful inputs regarding mouse behavioral assays and Dr. Mitsuharu Midorikawa for help with photo acquisition. We thank Mai Hatakenaka, Emiko Naraba, Tomoya Tsuchida, Sachie Sekino, Fumiya Urata and Yumi Tani for technical help. We greatly appreciate Takanori Maeda and Junnosuke Ohnuma for the effort in mouse behavioral analysis. We also thank Dr. Tomomi Shimogori for critically reading this manuscript prior to submission. We are grateful for the communications with Japan and international FOXG1 family communities. This work was supported by Grants-in-Aid for Scientific Research JP17H05775, JP17K07102, JP19H04789, JP19H05228, and JP20K07362 (G.M.), JP15H01667, JP19H03343, JP20H05481 and JP20H05916 (M.M.), and NIH grant R01MH095147 (G.F.). G.M. is supported by the Believe in a Cure, Inc., Mochida Memorial Foundation for Medical and Pharmaceutical Research, Cell Science Research Foundation, Takeda Science Foundation, and Brain Science Foundation.

## Author contributions

conceptualization (G.M. and R.M.), project administration and supervision (G.M. and M. M.), data curation, formal analysis, investigation and methodology (G.M., Y.U., A.N., K. H., H.O., Y. Yagasaki, and Y.K.), funding acquisition and resources (G.M., Y. Yanagawa, G.F., and M.M.), software (G.M., Y.U., A.N., and H.O.), validation and visualization (G. M., Y.U., A.N., K.H., and R.M.), writing original draft (G.M. and R.M.) and editing (G. M., G.F., R.M., and M.M.).

## Competing interests

The authors declare no competing interests.
