## [Peer Review File · Nature Communications]

Reviewers' Comments:

Reviewer #1:

Remarks to the Author:

In the present work Miyoshi and colleagues manipulate the expression level of Foxg1, a pivotal transcription factor during forebrain development and whose alterations in gene dosage are related to autistic spectrum disorder (ASD) symptomatology.

For this, they employ a complex and elegant strategy of genetic mouse breeding to obtain Foxg1 haploinsufficiency only in postmitotic neurons (both pyramidal and GABAergic) (Low model) or to overexpress it during specific time windows (High model). By testing those mice for behavioral traits associated with the autistic syndrome (sociability, anxiety and cognition), they highlight that haploinsufficiency of FoxG1 is enough to impair sociability in the three chamber test. This autistic feature is not depending on the role of FoxG1 in progenitor cells but in postmitotic neurons, and they appear only when Foxg1 is halved in both glutamatergic and GABAergic neurons. Similarly, in the High model, mice exhibit autistic features only when both Glutamatergic and GABAergic neurons were affected and upregulation in the 2nd week of age appears to be necessary and sufficient to induce ASD phenotype. Differently from the classic haploinsufficiency FoxG1 mouse model, both Low and High FoxG1 models do not show any defect in anxiety and cognition. Interestingly, an increase E/I balance reported in haploinsufficient FoxG1 models (but not in High model) in juvenile mice (2 weeks) but it is restored in adulthood. Gad2 depletion in Foxg1 het mice induce a worsening of ASD phenotype and GABAergic interneuron precursor transplantation is sufficient to ameliorate it.

The work is elaborated but well designed and the evidences provided are sound. Conclusions achieved would be of great interest for a better understanding of the neurodevelopmental disorders.

However, some points need to be clarified or better highlighted:

1) The effects of Foxg1 level increase on sociability defects in mice overexpressing Foxg1 in both Gluta and Gaba neurons are very mild (Figure 2B). Indeed, those mice do not show a preference for staying with the familiar mouse but when the stranger mouse is put in the right chamber (novelty), they clearly prefer that mouse. Accordingly, spectral power analysis in those mice also show a slight shift of the peak toward lower frequency and not a global reduction like in Low models. Finally, no alteration of E/I balance is reported in High model.

Therefore, it seems that the effects of FoxG1 overexpression is very different from those of haploinsufficient models. This should be better evidenced in the text.

Given this difference, the evidence that increased levels of FoxG1 in the second postnatal week is sufficient for induction of ASD phenotype cannot really be extended also to the haploinsufficiency, without any clear demonstration of this concept.

2) In Figure 3M the authors show that FoxG1 protein levels in the Low and High FoxG1 models are well analyzed by western blot analysis. In Figure 4, they exploit the same High FoxG1 model genetic tools to selectively overexpress FoxG1 in specific postnatal time windows. Considering the importance of this experiment for their conclusions, they need to check the exact levels of FoxG1 over the time in the different time windows, to assess how promptly FoxG1 levels are responding to the doxy administration. Also, the stability of FoxG1 protein may ensure a high level of the protein for few more days after doxy administration and this need to be checked and eventually considered in the determination of critical weeks.

3) How can the authors justify that although ASD phenotype emerge in Low models only when both Gaba and Gluta neurons have FoxG1 haploinsufficiency but GABAergic tone restoration is sufficient to rescue the phenotype? This should be better elaborated in the text.

Reviewer #2:

Remarks to the Author:

The manuscript titled "A FOXG1-dependent critical period for autism-associated GABAergic circuits" provides a comprehensive investigation of the role of the ASD related gene FOXG1 in

neurodevelopment and adult brain function. The experiments are well-designed, consisting of an impressive array of techniques to characterize the mouse model of ASD and the findings support a novel and critical role for FOXG1 in neuronal development and behaviors. The paper is generally well written and would be of interest to a broad scientific community. The authors are advised to address the following points to improve the manuscript.

1.(Fig 5C and D) The sociability and social novelty scoring in FoxG1 het/Gad2 null double mutant mice is pretty low, as compared to that in WT/Gad2 null mice. Given removal of the Gad2 gene reduces overall GABAergic function, although the authors argue that the FoxG1 heterozygous model develops ASD phenotypes through alterations in GABAergic systems, these data rather suggest the role of glutamatergic systems in the ASD phenotypes in FoxG1 heterozygous mice. The issues are need to be discussed.

2.(Fig 4D-G, 6C and D) I am unclear from the manuscript if the altered E/I balance is attributable to the impaired GABAergic systems. It would be helpful to provide raw data of the mEPSC and mIPSC frequencies and amplitudes.

3.(Fig 1A and C) More representative traces should be shown. For example, although, according to the traces, FoxG1 heterozygous mice spent almost in the middle area during social novelty sessions, these mice did not prefer the middle area (Fig 1C). The same is true for other figures showing representative behavioral traces.

4.(Typo, Figure 4 Legend, line 505) "Error bars" are mean +/- SEM.

Reviewer #3:

Remarks to the Author:

In this paper, the authors used FoxG1 mouse models to investigate the role of GABAergic neurons in the development of ASD phenotype, more specifically sociability and novelty recognition memory. They first showed that FoxG1 global heterozygous mice are impaired in sociability and recognition memory. Then they specifically deleted FoxG1 in GABAergic and excitatory neurons and showed that deletion in both types of neurons are required to produce the social deficits. Using the tetracycline inducible system, they found the overexpression of FoxG1 in both GABAergic and excitatory neurons also leads to sociability defects and alteration in EEG response. This also allowed them to define the developmental time window to postnatal 1-2 weeks, but not after, to be the critical time frame when manipulations of FoxG1 can lead to social defects that are associated with an increased E/I ratio. They also analyzed Gad2 KO mice and showed similar social deficits. In addition, Gad2KO/FoxG1 heterozygous mice showed worse phenotype than individual Gad2 KO or FoxG1 heterozygous mice. Finally, they showed that the transplantation of WT GABAergic precursor cells improves the social performance in FoxG1 hetero mice,

This is an important study. The identification of the time window to the first two postnatal weeks as the critical period when FoxG1 expression needs to be strictly regulated is of particular interest. The transplantation data are also interesting, suggesting a therapy strategy. Most of the experiments are done properly and the results support the claims. I have several comments that should be helpful to improve the paper.

Specific comments.

(1). It is clear that alterations in FoxG1 expression (either up or down) can lead to social deficits and E/I imbalance, but how this bidirectional change of FoxG1 results in these behavior and synaptic deficits remain unknown. In fact, it is not clear to me based on the E/I data, whether excitatory or inhibitory EPSCs or both are altered? What are the mechanisms underlying mEPSC/mIPSC changes? Is it presynaptic release or postsynaptic receptor? The data showed that both GABAergic and excitatory neurons need to be altered, but only the recording data in excitatory neurons are provided. In addition, the intrinsic properties of the neurons in both young and adult animals need to be analyzed because changes in these properties could also lead to social deficits.

(2). Can the time window identified in the FoxG1 model be generalized to other mouse models, at least to those with an altered E/I ratio?

(3). In Fig 2, representative images for Gad staining for the evaluation of the GABAergic neuronal number need to be provided. In addition, are these neuronal (Glu+GABA) heterozygous mice global? i.e., they are heterozygous in all Glu and GABA neurons in the entire brain? Some staining confirmation would be useful.

(4). In Fig 3 and 4, the use of the tetracycline inducible expression of FoxG1 is great and shows clear effects on sociability and E/I ratio. However, it is important to confirm the expression of the transgene, in both Glu and GABAergic neurons. Also is the expression global or in certain brain regions? These data would be important to interpret the results. Another important conclusion is that the overexpression of FoxG1 after three weeks has no effect on social behavior. In order to make such a conclusion, the expression level/patter of FoxG1 needs to be analyzed to ensure that they are similar to those in transgenic mice expressing FoxG1 around 1-2 weeks. As mentioned earlier, the intrinsic neuronal properties need to be analyzed in both young and adult mice in various models.

(5). In Fig. 4, I am not clear about how the E/I ratio was calculated. In addition, data on the frequency and amplitude of mEPSC and mIPSC should be provided separately to see if they are individually altered. Similar analysis in GABAergic neurons would be useful.

(6). In the GABAergic cell precursor transplantation experiments (Fig. 6), the number of transplanted cells/neurons needs to be quantified as compared to endogenous GABAergic neurons. In addition, the identity of these transplanted GFP+ cells needs to be confirmed. Again, for the recording data, individual mEPSP and mIPSC frequency and amplitude should be provided. Recording data from the transplanted cells/neurons are also useful. What happens when transplantation is done in adult mice? Although the transplantation approach is used here and shown to have an effect, whether the functional rescue (particularly in social behavior at 6 weeks) is due to restoration of GABA function specifically at 1 week or some other effects between 2-6 weeks is difficult to know. Experiments allowing acute restoration of GABA function during week 1 (e.g., using chemogenetics or allosteric modulators) may help.

Response to reviewers

We are tremendously grateful for the time and effort each of the reviewers put into reading our manuscript. Indeed, we feel their comments were very helpful in guiding us into improving our manuscript. All changes in the manuscript text file are shown by **blue color highlighting** besides the stylistic changes such as Figure numbers. Additional data we provide in the current version is as follows; 1) GABAergic cell precursor transplantation at 3weeks, in addition to the previous 1-week transplant attempts (Figures 6g to j). We found that carrying out cell transplantation during the critical period is essential for the Heterozygous animals to recover from the socially impaired ASD state. 2) Western blotting analyses in more details. We discovered that FoxG1 protein levels changes within two days upon addition or removal of the doxycycline-containing diet (Supplementary Figure 5). 3) Validation data for our cell transplantation experiments (Supplementary Figure 7). Basically, density of transplanted cells was comparable between the control wild type and Heterozygous cortex. 4) Intrinsic electrophysiological properties of pyramidal cells and putative fast-spiking interneurons were analyzed at P14 and in adults (Supplementary Figure 8). We found that the intrinsic properties of neurons are altered in the Heterozygous model. 5) We now provide complete data sets for the E/I ratio analyses by providing raw mEPSC and mIPSC data (Figures 4d-h, and Figures 6c-f).

Reviewer #1 (Remarks to the Author):

In the present work Miyoshi and colleagues manipulate the expression level of Foxg1, a pivotal transcription factor during forebrain development and whose alterations in gene dosage are related to autistic spectrum disorder (ASD) symptomatology.

For this, they employ a complex and elegant strategy of genetic mouse breeding to obtain Foxg1 haploinsufficiency only in postmitotic neurons (both pyramidal and GABAergic) (Low model) or to overexpress it during specific time windows (High model). By testing those mice for behavioral tracts associated with the autistic syndrome (sociability, anxiety and cognition), they highlight that haploinsufficiency of FoxG1 is enough to impair sociability in the three chamber test. This autistic feature is not depending on the role of FoxG1 in progenitor cells but in postmitotic neurons, and they appear only when Foxg1 is halved in both glutamatergic and GABAergic neurons. Similarly, in the High model, mice exhibit autistic features only when both Glutamatergic and GABAergic neurons where affected and upregulation in the 2nd week of age appears to be necessary and sufficient to induce ASD phenotype. Differently from the classic haploinsufficiency FoxG1 mouse model, both Low and High FoxG1 models do not show any defect in anxiety and cognition.

Interestingly, an increase E/I balance reported in haploinsufficient FoxG1 models (but not in High model) in juvenile mice (2 weeks) but it is restored in adulthood. Gad2 depletion in Foxg1 het mice induce a worsening of ASD phenotype and Gabaergic interneuron precursor transplantation is sufficient to

ameliorate it.

The work is elaborated but well designed and the evidences provided are sound. Conclusions achieved would be of great interest for a better understanding of the neurodevelopmental disorders.

However, some points need to be clarified or better highlighted:

1) The effects of Foxg1 level increase on sociability defects in mice overexpressing Foxg1 in both Gluta and Gaba neurons are very mild (Figure 2B). Indeed, those mice do not show a preference for staying with the familiar mouse but when the stranger mouse is put in the right chamber (novelty), they clearly prefer that mouse. Accordingly, spectral power analysis in those mice also show a slight shift of the peak toward lower frequency and not a global reduction like in Low models. Finally, no alteration of E/I balance is reported in High model. Therefore, it seems that the effects of FoxG1 overexpression is very different from those of haploinsufficient models. This should be better evidenced in the text. Given this difference, the evidence that increased levels of FoxG1 in the second postnatal week is sufficient for induction of ASD phenotype cannot really be extended also to the haploinsufficiency, without any clear demonstration of this concept.

- We agree with the reviewer that the High models show milder phenotypes for social novelty preference compared to the Het and Low models (Figure 3b). However, when we score sociability by combining sociability and social novelty assays (Figure 3c), High animals show comparable scores to the Het and Low (Lines 167-170). Based on these parameters, we think that the High model exhibits qualitatively distinct ASD phenotypes, which is consistent with the differences between patient cases of haploinsufficiency versus duplication of FOXG1. We now elaborate on this in the Discussion session (Lines 377-386), as well as the E/I ratio and EEG issues raised by the reviewer. We appreciate the reviewers concerns for the critical period argument based on the different High models. While excess copy number of FOXG1 has been identified in syndromic forms of ASD, the ectopic up-regulation of FOXG1 expression has been suggested to be a key contributor to disease etiology in idiopathic ASD patients (Mariani et al., 2015). We demonstrate that augmentation between 1 to 2 weeks is sufficient for the animals to develop a significant decrease in the E/I balance (amplitude; $p=3.61 \times 10^{-8}$, Figure 4e), an increase in gamma EEG power, and attenuation in sociability (but not social novelty) compared to the wild type (Figure 4b). Augmentation of FoxG1 expression starting from 3 weeks does not affect the EEG or social behavior. Thus, although we agree that the ASD phenotypes of the High model are qualitatively different from the Het and Low models, we think it is meaningful to address the issue of a critical period based on our High models with overlapping augmentation periods.

2) In Figure 3M the authors show that FoxG1 protein levels in the Low and High FoxG1 models are well analyzed by western blot analysis. In Figure 4, they exploit the same High FoxG1 model genetic tools to

selectively overexpress FoxG1 in specific postnatal time windows. Considering the importance of this experiment for their conclusions, they need to check the exact levels of FoxG1 over the time in the different time windows, to assess how promptly FoxG1 levels are responding to the doxy administration. Also, the stability of FoxG1 protein may ensure a high level of the protein for few more days after doxy administration and this need to be checked and eventually considered in the determination of critical weeks.

- A similar issue has been raised by the Reviewer 3 (Point 4). We now have carried out Western blotting analyses to identify the FoxG1 levels in the mPFC of the different High models of 1-2 week High (ASD-like sociability at 6 weeks) and 3 week High (Wild type-like sociability at 6 weeks) animals. We first confirmed that in the 1-2 week High model, removal of Dox at 1 week results in FoxG1 augmentation within a week (Figure 3o and Supplementary Figure 5d) to levels comparable to the much later timepoint (6 weeks). We further confirmed that the FoxG1 levels are already increased after two days of Dox removal (Supplementary Figure 5d), and decreased to levels comparable to the wild type animals after 2 days of Dox administration (Supplementary Figure 5d). For the 3 week High animals, we confirmed that FoxG1 is augmented to a similar extent to the 1 week High animals already after 2 days as well as at later in adults (Supplementary Figure 5d).

3) How can the authors justify that although ASD phenotype emerge in Low models only when both Gaba and Gluta neurons have FoxG1 haploinsufficiency but GABAergic tone restoration is sufficient to rescue the phenotype? This should be better elaborated in the text.

- We appreciate the reviewer for pointing this out. Although we still don't have mechanistic insights, it is clear that manipulating FoxG1 in both excitatory and inhibitory populations is necessary for the animals to develop ASD phenotypes. Our current hypothesis is that proper FoxG1 levels are necessary for the homeostatic machinery in neurons in general, both pre- and post-synaptically. In the medial PFC, if either excitatory or inhibitory cells maintain intact FoxG1 levels, the local circuits adjust to maintain function, which enables animals to develop normal social behaviors. We now have added our interpretation in the discussion session (lines 360 to 372).

Reviewer #2 (Remarks to the Author):

The manuscript titled "A FOXG1-dependent critical period for autism-associated GABAergic circuits" provides a comprehensive investigation of the role of the ASD related gene FOXG1 in neurodevelopment and adult brain function. The experiments are well-designed, consisting of an impressive array of techniques to characterize the mouse model of ASD and the findings support a novel and critical role for FOXG1 in neuronal development and behaviors. The paper is generally well written and would be of

interest to a broad scientific community. The authors are advised to address the following points to improve the manuscript.

1.(Fig 5C and D) The sociability and social novelty scoring in FoxG1 het/Gad2 null double mutant mice is pretty low, as compared to that in WT/Gad2 null mice. Given removal of the Gad2 gene reduces overall GABAergic function, although the authors argue that the FoxG1 heterozygous model develops ASD phenotypes through alterations in GABAergic systems, these data rather suggest the role of glutamatergic systems in the ASD phenotypes in FoxG1 heterozygous mice. The issues are need to be discussed.

- Following the Reviewer's suggestion, we now have included a paragraph in the discussion (Lines 404 to 410).

2.(Fig 4D-G, 6C and D) I am unclear from the manuscript if the altered E/I balance is attributable to the impaired GABAergic systems. It would be helpful to provide raw data of the mEPSC and mIPSC frequencies and amplitudes.

- Following the Reviewer's request, we now show the bar graphs for the mEPSC and mIPSC raw data.

3.(Fig 1A and C) More representative traces should be shown. For example, although, according to the traces, FoxG1 heterozygous mice spent almost in the middle area during social novelty sessions, these mice did not prefer the middle area (Fig 1C). The same is true for other figures showing representative behavioral traces.

- We very much appreciate the reviewer pointing this out and have now revisited the traces of the animals to include more representative trials. For example, in the Fig1a Hets, we now provide traces which show more representative behavior as well as times spent in each chamber that better illustrate the average data in the bar graphs.

4.(Typo, Figure 4 Legend, line 505) "Error bars" are mean +/- SEM.

- Thank you very much for kindly pointing this out.

Reviewer #3 (Remarks to the Author):

In this paper, the authors used FoxG1 mouse models to investigate the role of GABAergic neurons in the development of ASD phenotype, more specifically sociability and novelty recognition memory. They first

showed that FoxG1 global heterozygous mice are impaired in sociability and recognition memory. Then they specifically deleted FoxG1 in GABAergic and excitatory neurons and showed that deletion in both types of neurons are required to produce the social deficits. Using the tetracycline inducible system, they found the overexpression of FoxG1 in both GABAergic and excitatory neurons also leads to sociability defects and alteration in EEG response. This also allowed them to define the developmental time window to postnatal 1-2 weeks, but not after, to be the critical time frame when manipulations of FoxG1 can lead to social defects that are associated with an increased E/I ratio. They also analyzed Gad2 KO mice and showed similar social deficits. In addition, Gad2KO/FoxG1 heterozygous mice showed worse phenotype than individual Gad2 KO or FoxG1 heterozygous mice. Finally, they showed that the transplantation of WT GABAergic precursor cells improves the social performance in FoxG1 hetero mice,

This is an important study. The identification of the time window to the first two postnatal weeks as the critical period when FoxG1 expression needs to be strictly regulated is of particular interest. The transplantation data are also interesting, suggesting a therapy strategy. Most of the experiments are done properly and the results support the claims. I have several comments that should be helpful to improve the paper.

Specific comments.

(1). It is clear that alterations in FoxG1 expression (either up or down) can lead to social deficits and E/I imbalance, but how this bidirectional change of FoxG1 results in these behavior and synaptic deficits remain unknown. In fact, it is not clear to me based on the E/I data, whether excitatory or inhibitory EPSCs or both are altered? What are the mechanisms underlying mEPSC/mIPSC changes? Is it presynaptic release or postsynaptic receptor? The data showed that both GABAergic and excitatory neurons need to be altered, but only the recording data in excitatory neurons are provided. In addition, the intrinsic properties of the neurons in both young and adult animals need to be analyzed because changes in these properties could also lead to social deficits.

- We very much appreciate the reviewer's comments to improve the quality of our manuscript. We have now included raw data for mEPSC and mIPSC, in addition to the E/I ratio analysis in all of our studies (Figures 4d-h and 6c-f). Furthermore, we agree with the reviewer's suggestion to characterize the intrinsic properties of the cortical neurons. We now provide data from layer 2/3 pyramidal and putative fast-spiking interneurons at 2 weeks and 6 weeks of age in Supplementary Figure 8. Because of technical limitations, the whole-cell patch of inhibitory neurons based on GFP was only possible in the Heterozygous model, which we think most closely recapitulates the human disease condition. The FoxG1 Low and High animal models contain two Cre-lines for the excitatory and inhibitory populations (Nex-Cre, Dlx-Cre) and thus we could not visualize the

neuronal types in slices. We thank the reviewer for the helpful suggestion since we were able to uncover deficits in the intrinsic properties (Supplementary Figure 8).

(2). Can the time window identified in the FoxG1 model be generalized to other mouse models, at least to those with an altered E/I ratio?

- This is a very interesting but at the same time a tough question to answer. As far as we know, most E/I analyses have been done in mature animals (Antoine et al., 2019 Neuron, Feldman lab). Thus, we are also very curious to see the E/I analysis outcome in other established ASD models during the juvenile stages. We have now included a line in the discussion section to address this point (Lines 410 to 412).

(3). In Fig 2, representative images for Gad staining for the evaluation of the GABAergic neuronal number need to be provided. In addition, are these neuronal (Glu+GABA) heterozygous mice global? i.e., they are heterozygous in all Glu and GABA neurons in the entire brain? Some staining confirmation would be useful.

- We apologize that our presentation was not clear. We now have Gad1 *in situ* panels in the Figures 2d,e (control), Figure 2f,g (Het) and Figure 2h,i (Neuronal Het / Low). Neuronal heterozygous mice are NOT global, they are heterozygous in Glu and GABA neurons of the forebrain region (Figures 2c and 2l). We very much appreciate the comment since we did not clearly state that FoxG1 is selectively expressed in the forebrain region and not in other CNS populations such as hindbrain and spinal cord. We included those in lines 133 and 135. For the staining confirmation issue, since it is not easy to reliably quantify FoxG1 levels by immunohistochemistry, we carried out Western Blotting and found that FoxG1 levels of the Neuronal Het (with both Nex- and Dlx-Cre) is reduced to a similar extent as the global Heterozygous model at 2 and 6 weeks (Figure 3o).

(4). In Fig 3 and 4, the use of the tetracycline inducible expression of FoxG1 is great and shows clear effects on sociability and E/I ratio. However, it is important to confirm the expression of the transgene, in both Glu and GABAergic neurons. Also is the expression global or in certain brain regions? These data would be important to interpret the results. Another important conclusion is that the overexpression of FoxG1 after three weeks has no effect on social behavior. In order to make such a conclusion, the expression level/patter of FoxG1 needs to be analyzed to ensure that they are similar to those in transgenic mice expressing FoxG1 around 1-2 weeks. As mentioned earlier, the intrinsic neuronal properties need to be analyzed in both young and adult mice in various models.

- We again apologize that our presentation was not clear. In the tetracycline inducible High models, the Cre-lines we utilized are the forebrain-specific Nex-Cre (Excitatory) and Dlx-Cre (Inhibitory)

lines. Thus, FoxG1 GOF occurs not only in the cortex but includes regions such as striatum, hippocampus. We have now included a line in the results (line 163) and a paragraph in the discussion clearly stating that our manipulation is carried out not just in the cortex but in the forebrain regions (Lines 364 to 368). We agree with the reviewer that it is important to check the extent of FoxG1 increase in our 3 week High model, which shows no obvious social deficits at 6 weeks. In fact, this was also pointed out by the Reviewer 1 (second point) and we have now collected additional Western blotting data in Supplementary Figure 5 to resolve this issue. The intrinsic neuronal properties are now included in Supplementary Figure 8.

(5). In Fig. 4, I am not clear about how the E/I ratio was calculated. In addition, data on the frequency and amplitude of mEPSC and mIPSC should be provided separately to see if they are individually altered. Similar analysis in GABAergic neurons would be useful.

- We very much appreciate the comment by the reviewer. We now provide raw data for the mEPSC and mIPSC in all of our E/I ratio analyses (Figures 4d-h and 6c-f). As described above in the response to point (1), we unfortunately do not have a suitable strategy to visualize interneurons with GFP in our Low and High models as two Cre lines (Nex- and Dlx-Cre) are already simultaneously used in these animal models. We hope that the reviewer agrees with us that most studies on ASD-related mutant mice utilize layer 2/3 pyramidal cells for the E/I analysis (Antoine et al., 2019, and many others).

(6). In the GABAergic cell precursor transplantation experiments (Fig. 6), the number of transplanted cells/neurons needs to be quantified as compared to endogenous GABAergic neurons. In addition, the identity of these transplanted GFP+ cells needs to be confirmed. Again, for the recording data, individual mEPSP and mIPSC frequency and amplitude should be provided. Recording data from the transplanted cells/neurons are also useful. What happens when transplantation is done in adult mice? Although the transplantation approach is used here and shown to have an effect, whether the functional rescue (particularly in social behavior at 6 weeks) is due to restoration of GABA function specifically at 1 week or some other effects between 2-6 weeks is difficult to know. Experiments allowing acute restoration of GABA function during week 1 (e.g., using chemogenetics or allosteric modulators) may help.

- We now have quantified the numbers of the transplanted GABAergic cells and compared to the numbers of the endogenous GABAergic neurons. We took advantage of GFP expression of the donor GABAergic cells and combined with Gad1 in situ hybridization visualizing the overall GABAergic cell numbers (Supplementary Figure 7b-d). For the recording data, in a similar manner to the other E/I ratio panels, we now provide raw data for mEPSC and mIPSC (Figures 6c-f). We very much appreciate the reviewer's suggestion to test if cell transplantation at 1 week is necessary to ameliorate the sociability deficits, rather than at between 2-6 weeks. Following

this comment, we further carried out another set of cell transplantation experiments at a later timepoint of 3 weeks (Figures 6i and 6j) and found out that this is not effective (Supplementary Figure 7e, f). We strongly agree with the reviewer that inclusion of this new data has improved the significance of our manuscript by helping define the critical time-window for cell transplantation. We also agree with the reviewer that it would be very interesting to test the effect of GABA receptor agonists at 1 week of age and in fact we have tried such experiments. In our hands, it was technically challenging to utilize infusion pumps in 1 week old pups and unfortunately, we were not able to make the pups survive. We hope that in the future we may be able to address this interesting point experimentally.

Reviewers' Comments:

Reviewer #1:

Remarks to the Author:

The authors have properly addressed all queries of this reviewer.

Reviewer #2:

Remarks to the Author:

All my concerns are satisfactorily addressed in the revised manuscript.

Reviewer #3:

Remarks to the Author:

This is a much improved manuscript. The authors have made impressive effort and addressed my concerns. I have a few minor comments.

I agree that the altered E/I balance may be the key reason to result in behavior deficits, but how this altered E/I balance of synaptic transmission is (or not) related to altered intrinsic neuronal excitability and whether the altered excitability independently contributed to behavior defects remains unknown. These issues can be discussed.

Also, I agree that altered E/I ratio is clear in these models, but whether this is mainly caused by altered GABAergic function is not very clear to me as evidenced in Fig 4d and e, where mEPSC changes appear to be more drastic than mIPSCs. Although I do agree that decreasing GABA function made the phenotype worse and increasing GABA function could rescue the phenotype, this does not mean that the underlying cause is GABAergic. Caution must be taken to interpret the data.

Point-by-point response to the reviewers' comments.

We would like to thank all of the reviewers for their time and effort and also would like to appreciate for their quick response to our revised materials.

Reviewer #1 (Remarks to the Author):

The authors have properly addressed all queries of this reviewer.

Reviewer #2 (Remarks to the Author):

All my concerns are satisfactorily addressed in the revised manuscript.

Reviewer #3 (Remarks to the Author):

This is a much improved manuscript. The authors have made impressive effort and addressed my concerns. I have a few minor comments.

I agree that the altered E/I balance may be the key reason to result in behavior deficits, but how this altered E/I balance of synaptic transmission is (or not) related to altered intrinsic neuronal excitability and whether the altered excitability independently contributed to behavior defects remains unknown. These issues can be discussed.

Also, I agree that altered E/I ratio is clear in these models, but whether this is mainly caused by altered GABAergic function is not very clear to me as evidenced in Fig 4d and e, where mEPSC changes appear to be more drastic than mIPSCs. Although I do agree that decreasing GABA function made the phenotype worse and increasing GABA function could rescue the phenotype, this does not mean that the underlying cause is GABAergic. Caution must be taken to interpret the data.

- We appreciate the reviewer's evaluation of our manuscript. We agree with the reviewer's point of '*whether the altered excitability independently contributed to behavior defects remains unknown*'. We added few lines in the discussion to describe the results of intrinsic firing property analyses (in red fonts).